# The Fundamental Role of Oxime and Oxime Ether Moieties in Improving the Physicochemical and Anticancer Properties of Structurally Diverse Scaffolds

**DOI:** 10.3390/ijms242316854

**Published:** 2023-11-28

**Authors:** Jean Fotie, Caitlyn M. Matherne, Jasmine B. Mather, Jordan E. Wroblewski, Khaitlynn Johnson, Lara G. Boudreaux, Alba A. Perez

**Affiliations:** Department of Chemistry and Physics, Southeastern Louisiana University, SLU 10878, Hammond, LA 70402-0878, USA; caitlyn.matherne@selu.edu (C.M.M.); jasmine.mather@selu.edu (J.B.M.); jordan.wroblewski@selu.edu (J.E.W.); khaitlynn.johnson@selu.edu (K.J.); lara.boudreaux@selu.edu (L.G.B.); alba.perezanchel@selu.edu (A.A.P.)

**Keywords:** oximes, oxime ethers, anticancer activities, physicochemical properties, SAR, docking studies

## Abstract

The present review explores the critical role of oxime and oxime ether moieties in enhancing the physicochemical and anticancer properties of structurally diverse molecular frameworks. Specific examples are carefully selected to illustrate the distinct contributions of these functional groups to general strategies for molecular design, modulation of biological activities, computational modeling, and structure–activity relationship studies. An extensive literature search was conducted across three databases, including PubMed, Google Scholar, and Scifinder, enabling us to create one of the most comprehensive overviews of how oximes and oxime ethers impact antitumor activities within a wide range of structural frameworks. This search focused on various combinations of keywords or their synonyms, related to the anticancer activity of oximes and oxime ethers, structure–activity relationships, mechanism of action, as well as molecular dynamics and docking studies. Each article was evaluated based on its scientific merit and the depth of the study, resulting in 268 cited references and more than 336 illustrative chemical structures carefully selected to support this analysis. As many previous reviews focus on one subclass of this extensive family of compounds, this report represents one of the rare and fully comprehensive assessments of the anticancer potential of this group of molecules across diverse molecular scaffolds.

## 1. Introduction

Cancer is a generic term used to describe a broad spectrum of diseases characterized by the rapid formation of abnormal cells that grow uncontrollably, extending beyond their usual boundaries and invading adjacent parts of the body while destroying normal tissues [1,2,3]. This condition can affect any part of the body, with the 2020 estimates from the World Health Organization (WHO) indicating that the most common types in terms of new cases were breast cancer (2.26 million cases), lung cancer (2.21 million cases), colon and rectal cancer (1.93 million cases), prostate cancer (1.41 million cases), nonmelanoma skin cancer (1.20 million cases), and stomach cancer (1.09 million cases), with the total number of new cases estimated to 19.3 million [4,5,6,7]. In fact, as mortality rates for other leading causes of death such as stroke and coronary heart disease decline in many parts of the world, cancer is emerging as the deadliest disease worldwide, and according to the WHO estimates, cancer ranks as the first or second leading cause of death before the age of 70 years in 112 out of 183 countries, and as the third or fourth leading cause in about 23 other countries [4,5,6,7,8,9]. In 2020, cancer was responsible for nearly one in six deaths, accounting for nearly 10 million deaths globally [4,5,6,7]. Compounding the issue, the overall cancer hardship across the globe is expected to reach 28.4 million cases by 2040, representing a 47% increase from 2020 [4,10,11]. One unique characteristic of cancer, unlike many other diseases, is its higher overall incidence in developed countries compared to transitioning or low-income countries, with rates ranging from 2 to 3 times higher [4,9,11,12]. This situation is further worsened by the prevailing trend of resistance to numerous available therapies, added to their pervasive side effects [13,14,15,16,17].

Fortunately, novel approaches to cancer treatment through a surgical targeting of tumors’ microenvironment, such as longitudinal single-cell profiling of chemotherapy [18,19] or targeted immunotherapies, including chimeric antigen receptor (CAR) cell therapy [20,21], immune checkpoint inhibitors [22,23,24], monoclonal antibodies [25,26,27], and immune system modulators [28,29,30], are demonstrating significant promise across various types of cancers. Nevertheless, small molecules have always been the bedrock of cancer therapy and are predicted to continue playing a critical role in the future. One such class of small molecules is oximes.

Oximes are nitrogen-containing molecules from the imine family, characterized by the general formula (RR′C=N–OH). They are available in the form of aldoximes or ketoximes, and have been well-known since the 1960s to be widely distributed across various species in all realms of life [31,32], with the first synthetic member of the family (methylglyoxime, **1**) reported in 1882 by Meyer and Janny [33]. In plants, they are viewed as critical metabolic bifurcation points between general and specialized pathways, with the majority of plant oximes originating from amino acids through processes catalyzed by the cytochrome P450 family of enzymes [31,32]. As a result, the structures of naturally occurring oximes are not very complex, as they mirror their parent amino acids. This can be observed with the first oximes from plant origin, namely isobutyraldoxime (**2**) [34] (derived from valine) and phenylacetaldoxime (**3**) [35] (derived from phenylalanine), which were identified in 1967 from *Linum usitatissimum* and *Tropaeolum majus*, respectively (Figure 1). Additionally, indole-3-acetaldehyde oxime (**4**) (derived from *L*-tryptophan) was isolated from *Brassica oleracea* just a year later [36]. In fact, plant metabolites such as auxin, cyanogenic glucosides, glucosinolates, and other bioactive volatile compounds were also shown to originate from oximes [32,36,37]. This family of compounds and their derivatives has also been suggested to play key roles in growth regulation, plant defense, pollinator attraction, and plant communication with the surrounding environment [38,39,40]. Furthermore, compounds derived from oximes can serve as quenchers for reactive oxygen species in plants or act as storage compounds for reduced nitrogen, which may be released on demand through the activation of endogenous turnover pathways [31,32,41].

As highly bioactive molecules, both naturally occurring and synthesized oximes have found versatile uses in many spheres of society, particularly in the agricultural and medical sectors. These hydroxy-imine derivatives are well-known for their antibacterial, antifungal, anti-inflammatory, antioxidant, and anticancer activities [42,43,44,45,46,47]. A recent review by Dhuguru et al. [48] provides extensive insights into the synthesis, mechanism of action, and pharmacokinetics of oxime-based FDA-approved drugs. The anticancer activity of this family of compounds is also well-documented, with several notable reviews published on the topic in recent years [45,49,50,51,52,53]. However, each of these reports has primarily focused on one subclass of this extensive family of compounds, thus failing to provide a clear view of the diversity of scaffolds encompassed within this group of molecules. The present review delves into the fundamental role of oxime and oxime ether moieties in enhancing the physicochemical and anticancer properties of a structurally diverse molecular framework. Specific selected examples are used to illustrate the distinct involvement of the oxime functional group in general strategies for molecular design, modulation of biological activities, computational modeling, and structure−activity relationships studies. 

## 2. Classes of Anticancer Oximes

### 2.1. Anticancer Activity of Indole Based-Oximes and Oxime Ethers

Among the indole-based oximes and oxime ethers, derivatives of indirubin (**5**) have been extensively investigated for their anticancer activity. This purplish-red bisindole alkaloid known as the key active ingredient in traditional Chinese medicine herbal formulas, has been effective in treating human acute promyelocytic leukemia (APL) [54,55,56,57]. In fact, this Chinese concoction composed of *Radix psudostellariae* and *Salvia miltiorrhiza*, has demonstrated compelling synergistic effects in inducing differentiation of acute promyelocytic leukemia cells in vitro. Additionally, it has shown significant therapeutic activity in murine acute promyelocytic leukemia animal models [58,59] and thus has advanced into clinical trials for treating childhood acute promyeloid leukemia in China [55,57].

Numerous studies have also indicated that indirubin possesses a strong affinity for cyclin-dependent kinases (CDKs), which play an essential role in controlling cell cycle and proliferation [60,61]. This molecule was also shown to interact with glycogen synthase kinase-3β (GSK-3β) by interfering with the ATP-binding site [62], with its anticancer activity arising through the induction of cell cycle arrest in the G1 or G2 phase of tumor progression, leading to apoptosis in various cancer cell types [60,63,64]. While CDKs are key enzymes governing cell cycle progression [65], GSK-3β has been reported to play important roles in transcription by regulating the activities of key factors, such as NFκB [66], transcriptional factor EB [67], and in cell signaling pathways, such as growth factor signaling [68] and Wnt signaling [69].

However, indirubin’s therapeutic effectiveness is hindered by its poor water solubility, inadequate pharmacokinetic properties, low absorption rate, and significant gastrointestinal toxicity [55,70]. These factors pose substantial obstacles to its clinical application. Consequently, multiple strategies have been employed to enhance the pharmacodynamic and pharmacokinetic properties of this family of compounds. One such strategy involves transforming the carbonyls from the indirubin’s scaffold into oxime or oxime ether pharmacophores, as illustrated in Figure 2. This approach has led to the preparation of a wide variety of indirubin C-3′-oxime (**6**) derivatives, many of which exhibited superior properties compared to the naturally occurring parent alkaloid, including enhanced water solubility and better selectivity [55,61,70,71,72]. For example, while exploring the anticancer activity of a series of *N*^1^-alkylindirubin C-3′-oximes against MCF-7, LOVO, and LNCAP cells, Wang et al. [73] showed that many of these derivatives exhibited moderate cytotoxicity, with **7** being the most active across all these cell lines, displaying IC_50_ values ranging from 6.72 to 13.38 μM.

The effects of various substitution patterns on the activity of these compounds have also been extensively investigated. In fact, Lee et al. [74] demonstrated that indirubin-5-nitro-3′-oxime (**8**) exhibits superior antiproliferative activity against A549, HT-1080, and HL-60 cells compared to indirubin-3′-oxime, with IC_50_ values of 5.4, 5.9, and 9.2 µM, respectively. Further studies have indicated that the presence of an electron-withdrawing group at the 5-position enhances the anticancer activity of these compounds. Out of the 29 molecules synthesized and evaluated for their effects on MV4-11 cells (with FLT3/ITD mutation), compound **9** displayed the highest inhibition potency, with an impressive IC_50_ value of 0.072 μM [75]. Moreover, this molecule was found to selectively inhibit the activity of FLT3 kinase and induce cell cycle arrest at the G0/G1 phase in MV4-11 cells [75]. Additional structure–activity relationship studies revealed that a sulfonate group at the 5-position improved the inhibitory activity against FLT3 kinase. However, this substitution was found to be detrimental to antiproliferative activity due to the resulting molecule’s low cell membrane permeability [75]. Nevertheless, 5-methoxyindirubin 3′-oxime (**10**) has been demonstrated to inhibit PDAC cell proliferation, with the administration of compound **10** leading to the inhibition of PDAC xenograft growth in BALB/c nu/nu mice, and further studies suggesting that this analog might exert its activity by inducing G2/M phase arrest in PDAC cells through the inhibition of CDK1/cyclin B1 levels, subsequently leading to apoptosis [76]. Furthermore, 5-diphenylacetamido-indirubin-3′-oxime (**11**), a mitochondria-targeting antileukemic agent, exhibited strong inhibitory activity against a panel of drug-sensitive and drug-resistant primary and malignant leukemia cell lines [77]. This compound rapidly collapses the mitochondrial membrane potential (MMP), followed by the release of cytochrome c into the cytosol, and a severe depletion of cellular ATP [77]. These findings indicate that this molecule induces leukemia cell death through a mechanism distinct from that of many other indirubin analogs [77].

Death-associated protein (DAP) kinases, including DRAK2 (apoptosis-inducing kinase 2), are well known to perform apoptosis-related tasks in cells [78]. A number of studies have shown that DRAK2 is engaged in regulating the activation of T lymphocytes as well as pancreatic β-cell apoptosis in type I diabetes [78]. This kinase also plays an important role in the development of tumor-related diseases via diverse mechanisms, and has been shown to be involved in the exacerbation of alcoholic fatty liver disease (NAFLD) through SRSF6-associated RNA selective splicing, the regulation of chronic lymphocytic leukemia and acute myeloid leukemia, as well as the progression of colorectal cancer [78].

A number of 5-amido-indirubin-3′-oxime derivatives have been shown to exhibit potent inhibitory activity against DRAK1 and DRAK2 [79], with further structure–activity relationship studies indicating that small aliphatic amide at the 5-position were better promotor of the inhibitory activity against DRAK2. Among them, **12** (IC_50_ 3 nM and 51 nM), **13** (IC_50_ 14 nM and 24 nM), **14** (IC_50_ 14 nM and 120 nM), and **15** (IC_50_ 8 nM and 21 nM) displayed strong inhibitory activity against DRAK1 and DRAK2, respectively [79], with the molecular docking study revealing that compound **12** binds to the ATP-binding site in DRAK2 by forming three key H-bond interactions, specifically with Glu111, Ala113, and Glu117 [79].

Indirubin-3′-oxime analogs with carboxylic acid, ester, carbamide or sulfonamide at the 5-position have also been studied for their anticancer activity. In fact, a 5-methyl acetate analog of indirubin-3′-oxime (**16**) demonstrated potent inhibition of FLT3 kinase activity in vitro, with an IC_50_ of 7.89 nM [80]. In contrast, the corresponding free carboxylic acid (**17**) analog showed relatively weak activity against FLT3 (IC_50_ of 3.19 μM) [80]. Interestingly, only the 5-methyl acetate analog exhibited a strong cytotoxic effect against MV4-11 cells, while the carboxylic acid analog was inactive. Further studies revealed that compound **16** inhibited the phosphorylation of downstream STAT5, promoted PARP cleavage in MV4-11 cells, and induced apoptosis and cell cycle arrest in the G1 phase [80]. On the other hand, Merz and coworkers designed and synthesized a series of indirubin-5-carboxamide oxime derivatives with improved water solubility. These compounds were evaluated for their antiproliferative activity against LXFL529L cells, with **18** and **19** exhibiting noticeable inhibitory activity [81].

While evaluating the cytotoxicity of a panel of 5-sulfonamide indirubin oxime analogs with enhanced water solubility and improved physicochemical and pharmacological properties, Jautelat et al. [82] indicated that many of these analogs possess potent inhibitory activity toward CDK2. These molecules also exhibited significant antitumor activity against MCF-7 cells, with compounds **20** (IC_50_ 0.04 and 0.4 μM) and **21** (IC_50_ 0.04 and 0.1 μM) being the most active in both CDK2 inhibitory activity assay and tumor growth inhibition assay against MCF-7 cells [82].

The effects of other substitution patterns on the anticancer activity of this family of compounds have also been explored. 6-bromoindirubin-3′-oxime (**22**) was shown to inhibit the activity of glycogen synthase kinase 3β (GSK3β) [83]. This compound suppresses the proliferation, invasion, and migration of ovarian cancer cells (A2780 and OVCAR3), reduces lamellipodia formation, and induces G1 arrest of the cell cycle [83]. Exposure of these cells to compound **22** led to a significant downregulation of mRNA and protein expression of cyclin D1 and MMP9 compared with untreated control cells. Moreover, 6-bromoindirubin-3′-oxime strongly reduces the formation of lung metastasis in the well-established 4T1 mouse model of aggressive breast cancer [83]. Subtoxic concentrations of this analog affect a number of key characteristics of the metastatic process, including the inhibition of adhesion, migration, and invasion of a variety of metastatic cell types in vitro [83]. Surprisingly, RNAi-mediated silencing of glycogen synthase kinase 3β and phosphoinositide-dependent protein kinase 1 (PDK1), both modulators of cellular metastasis targeted by 6-bromoindirubin-3′-oxime, did not affect invasive migration in this study [83]. Instead, the Jak/STAT3 signaling pathway appeared to play a major role by modulating downstream migration regulators such as C-terminal tensin-like protein and matrix metalloproteinase 2 [83]. However, PDK1 and GSK3β still contributed to the overall response to 6-bromoindirubin-3′-oxime, as silencing all three pathways resulted in almost a complete inhibition of migration, mimicking the response of compound **22** [83].

In contrast to its 5-bromo and 6-bromo isomers, 7-bromoindirubin-3′-oxime (**23**) exhibited only marginal inhibitory activity towards CDKs and GSK-3 [71,84,85]. However, this latter analog induced the appearance of large pyknotic nuclei, without exhibiting classical features of apoptosis such as chromatin condensation and nuclear fragmentation. The induced cell death was not accompanied by cytochrome c release or any measurable effector caspase activation [71,84]. This result indicates that, unlike other indirubin analogs, 7-bromoindirubin-3′-oxime triggers the activation of a nonapoptotic cell death pathway, possibly through necroptosis or autophagy. In this study however, the 7-fluoro derivative (**24**) appeared to have better inhibitory activity towards CDKs and GSK-3 than its 7-bromo counterpart [71,84]. Nevertheless, 7-bromoindirubin-3′-oxime significantly reduced cell viability in 14 thyroid carcinoma cell lines, with treated cells showing DNA fragmentation, cell cycle arrest, and lactate dehydrogenase release, but no LC3B cleavage [85]. In other words, this compound induced a nonclassical kind of cell death that was caspase-independent and did not involve DNA fragmentation [85].

A number of studies focusing on the investigation of the anticancer activities of indirubin-3′-oxime bearing a diversity of substitution patterns on the bisindole scaffold have also been reported. In a comparative analysis against indirubin-3′-oxime parent compound, Ichimaru et al. [86,87] prepared and evaluated a series of analogs with a variety of bromo- or dibromo-substituted patterns as illustrated by compounds **25**–**30** in Figure 2. This variation in substitution patterns did not appear to significantly affect the activity of the resulting compounds, with IC_50_ values ranging from 4.4 to 14 μM [86,87].

While exploring other substitution patterns, Kim and coworkers demonstrated the 5,5′-substituted derivatives of indirubin-3′-oxime are potent CDK inhibitors, as many of these compounds displayed potent inhibitory effects on CDK2, with over 90% inhibition potency at 1 μM concentration [88,89,90]. 5′-Hydroxy-5-nitro-indirubin oxime (**31**) and 5′-fluoro-5-nitro-indirubin oxime (**32**) not only displayed selective and significant inhibitory activities towards CDK1 and CDK2, but also exhibited good anticancer activity against several cancer cell lines [88,90]. Compound **31** was able to inhibit the phosphorylation of the retinoblastoma protein (Rb), which is a major substrate of CDK, as well as CDK2/cyclin E activity (IC_50_ = 2.16 nM) more efficiently when compared to its effect on the CDK1/cyclin B activity (IC_50_ = 13.8 nM) [75,89,90]. Docking studies indicated that compound **31** formed a new hydrogen between the 5′-hydroxy and Asp86 in the solvent-accessible region of CDK2, with the 3′-oxime moiety creating a hydrogen bond with Ile10 instead of Gln131 [75,88]. Further SAR studies revealed that small substituents at the 5′-position, such as -OH or -F, as well as electron withdrawing substituents at the 5-position, such as -NO_2_, significantly increased the inhibition potency against CDK2 [75,88,90]. 

Compounds **31** and **32** were able to induce apoptosis in Imatinib-resistant chronic myeloid leukemia cells, as well as efficiently decrease the viability of CML-derived drug-sensitive and Imatinib-resistant K562 cells, both in vitro and in vivo [90]. Additionally, Park and co-workers indicated that compound **31** significantly decreased the viability of leukemia K562 cells with an IC_50_ value of 669 nM, while also inhibiting Imatinib-resistant K562R cells with a similar IC_50_ value of 783 nM [90]. A combination of **31** and Imatinib also appeared to increase the number of apoptotic K562R cells compared to the effect of either **31** or Imatinib, with the former (**31**, 15 mg/kg, but not Imatinib) significantly inhibiting the growth of K562R tumor cells in vivo [88,90,91].

Using the knowledge gathered from the elucidation of the X-ray structure of 5-bromoindirubin and CDK2, Yan et al. [92], prepared various 5′,6′-difluoro-indirubin derivatives, among which compounds **33**–**35** showed potent inhibitory effects on both CDK2/cyclin E1 and CDK9/cyclin T1 at nanomolar or low micromolar levels. Docking studies indicated that compound **35** positions well into the ATP pocket and exhibited similar bindings with hinge region residues in CDK9 and CDK2, with this 5-chloro analog unexpectedly forming two additional halogen bonds with Lys48 and Asp167 residues in CDK9, whereas, such a halogen bond was absent in CDK2, indicating that the selectivity exhibited by **35** towards CDK9 might be related to the formation of these halogen bonds. Yet, while compounds **33** and **34** showed a significant cytotoxic activity against five human cancer cell lines with IC_50_ values at low micromolar levels, **35** displayed only a marginal activity, a situation that might be attributable to its poor solubility [92]. 

Dual-specificity tyrosine phosphorylation-regulated kinases (DYRKs) are well known to be involved in multiple cellular functions, including intracellular signaling, mRNA splicing, chromatin transcription, DNA damage repair, cell survival, cell cycle control, differentiation, endocytosis, and synaptic plasticity [93], to name just a few. Myrianthopoulos et al. reported the synthesis of indirubin-3′-oxime-based selective DYRK inhibitors [94]. They revealed that a carboxylate substitution at the 5′-position in combination with a bromine at the 7-position significantly promoted selective DYRK inhibition, as illustrated with compounds **36** (IC_50_ 0.31 and 0.35 μM) and **37** (IC_50_ 0.21 and 0.13 μM) [94].

To rationalize the observed affinity and selectivity profiles, docking studies of compound **37** to a DYRK1a-derived homology model of DYRK2 revealed the existence of two distinct binding modes characterized by a 180° flip of the indirubin core in relation to either its primary or secondary axis [94]. In both cases, the typical hydrogen bond triplet between the indirubin pharmacophore and the kinase hinge was either disrupted (mode I) or inverted (mode II), with the 5′-carboxylate forming a salt bridge either with Lys165 (mode I) or with the catalytic Lys178 (mode II) [94]. This binding mode appears to be different from the experimental indirubin-kinase binding established in previous studies [95].

The crystal structure of **37** bound to DYRK2 (Figure 3) at a resolution of 2.28 Å confirmed that the binding mode was indeed inverted in comparison to previously reported indirubin-kinase complexes geometric arrangement [95], and consistent with the predicted binding mode II [94]. In fact, not only the crystal structure revealed that the predicted salt bridge anchoring the anionic 5′-carboxylate to the sidechain of Lys178, but also disclosed an additional hydrogen bond between the 5′-carboxylate and the NH backbone of Asp295, with three water molecules participating in the hydrogen bond network by bridging the inhibitor carboxylate and Lys178 ammonium groups to the sidechains of Asp295 and Glu193 [94].

Despite the numerous substitutions performed on the core bisindole skeleton, indirubin C-3′-oxime derivatives still exhibited solubility issues, particularly in vivo within live tissue. These poor solubility and inadequate pharmacokinetic properties are primarily attributed to the high polarity and hydrophilic nature of the oxime functional group. To modulate the hydrophilic/hydrophobic balance of this family of compounds, various substituents, including glycosides, polyhydric alcohols, epoxides, amide chains or their corresponding salts, were added to the 3′-oxime functional group, effectively transforming them into an oxime ether.

As such, Nam et al. [96] synthesized several indirubin-3′-oxime ethers by introducing diverse polyether or polyhydric alcohol chains to the hydroxyl group of the 3′-oxime. They demonstrated that compounds **38** and **39** strongly inhibited Stat3 signaling in four human prostate and breast cancer cell lines [96]. Additionally, **39** was found to directly inhibit c-Src kinase activity in vitro (IC_50_ = 0.43 μM) [96]. Treatment with the same compound appeared to suppress the tyrosyl phosphorylation of c-Src kinase’s downstream target protein Stat3, while concurrently downregulating the Stat3-associated proteins Mcl-1 and Survivin, triggering MDA-MB-468 breast cancer cell apoptosis [96].

Owing to the fact that many epoxides can be further metabolized by epoxide hydrolase in the cellular metabolic environment to generate the corresponding polyhydric alcohols, Miyairi and co-workers prepared a series of indirubin-3′-(O-oxiran-2-ylmethyl)-oxime derivatives designed to enhance cell permeability, and thus serving as prodrugs [86,87]. The anticancer activity of each molecule was evaluated on HepG2 cells through a comparative analysis against their respective indirubin-3′-oxime parent compounds with appropriate substituents [76,86,87]. This study indicated that all indirubin 3′-(O-oxiran-2-yl)oxime derivatives consistently outperformed their corresponding indirubin-3′-oxime intermediates [86,87].

A docking study with **40** at the ATP-binding site of GSK-3β designed to evaluate the positioning of this molecule within the site, including the interactions between the oxirane and amino acid residues, suggested that the oxirane group of **40** is positioned closely enough to Cys-199 to form a covalent linkage between them [86,87]. It should be mentioned that Cys-199 is a critical amino acid for GSK-3β kinase activity [62]. Given that halogen atoms in organic compounds have exhibited strong affinity for the sulfur atoms in cysteine and methionine residues in a number of proteins [97], the participation of Cys-199 in enhancing binding affinity through sulfur-halogen interaction was anticipated. The authors concluded that the sulfur-halogen interaction forces the oxirane moiety close to Cys-199, thereby accelerating the covalent bond formation between the oxirane and the thiol [76,86,87]. In addition, Kurita et al. [98] demonstrated that these indirubin-3′-(O-oxiran-2-ylmethyl)-oxime derivatives possess potent cytotoxicity against human neuroblastoma IMR-32 and SK-N-SH cell lines, with IC_50_ values of 0.16 and 0.07 µM, respectively. These molecules induce caspase-independent apoptosis by inhibiting DNA repair and causing DNA fragmentation in IMR-32 cells [76,98].

While exploring the effect of different substitution patterns on the behavior of these analogs, Ichimaru et al. [86,87] found that many indirubin-3′-oxime derivatives were potential anticancer agents, with compounds **41**–**44** exhibiting significant inhibitory activity against HepG2 cells, with IC_50_ values of 0.62, 1.7, 1.3, and 1.6 μM, respectively [86]. Notably, compound **41** (with a 6-bromo substituent) displayed the most promising anticancer profile, whereas the activity of compounds **42** and **44** was not superior to that of their corresponding unsubstituted analog **40** [86,87,98].

Further investigations by other research groups have indicated that compound **39** possesses a broad range of in vitro and in vivo activities against numerous cancer cell lines. For instance, **39** was found to significantly inhibit the growth of CT-26 allografts in syngeneic BALB/c mice without affecting body weight [99,100]. Immunohistochemistry revealed that **39** could induce apoptosis and hinder tumor angiogenesis, leading to substantial impediment of cell proliferation, migration, and tube formation in vascular endothelial growth factor (VEGF)-treated HUVECs [99,100]. In vivo studies showed that this analog has the potential to inhibit the formation of subintestinal vessels in a zebrafish model in a dose-dependent manner [100]. In a different study, Nam et al. [101] further indicated that **39** significantly blocked tyrosyl phosphorylation of Stat5, while inhibiting Stat5 DNA-binding activity in human K562, KCL-22M, and primary CML cell lines, with treatments with **39** strongly decreasing the autophosphorylation of Src and SFKs in K562 and KCL-22M cells at 5 μM, and in primary CML cells. This compound also induced apoptosis in chronic myelogenous leukemia cells by downregulating the expression levels of Stat5 target proteins Mcl-1 and Bcl-xL, highlighting the potential of **39** to inhibit SFK/Stat5 signaling in leukemia cells [101]. During an investigation into the effects of different substitution patterns on the anticancer activity of compound **39**, Cheng et al. [102] noted that compound **45**, a 5-methoxy derivative of **39**, could significantly block TGFb/BMP signaling by inducing ubiquitin-proteasome-mediated depletion of total R-Smads. This molecule also exhibited a strong inhibition of cell viability across several pancreatic cancer cell lines (Panc-1, MIA-PaCa2, BxPC3, and AsPC1), with IC_50_ values ranging from 0.76 to 2.2 μM [103]. 

Nevertheless, while further investigating the physicochemical and cell permeability properties of these molecules, Heshmati et al. [104] demonstrated that water solubility remained a significant hurdle for drug permeation, although a Calcein-AM uptake assay also indicated that **39** was not a good substrate for P-glycoprotein, suggesting that this compound could be a promising candidate for oral delivery due to its high membrane permeability [104]. It is worth mentioning that Jakobs et al. [64,105], while investigating some glycoside derivatives of indirubin C-3′-oxime, showed that **39**, **46**, and **47** exhibited potent cytotoxicity effects against LXFL529L, MCF-7, and HCT-116 cell lines at low micromolar concentrations [105], with compound **46** proven to block pRb phosphorylation in LXFL-529L cells [64].

Another strategy used to improve the physicochemical properties of this family of compounds is to attach a quaternary ammonium group to the oxime moiety. In line with this approach, Ginzinger et al. [106] prepared a series of carnitine oxime ether and choline oxime ether prodrugs. They not only investigated the activities of these compounds against three human cancer cell lines (A549, CH1, and SW480), but also assessed their suitability for clinical administration [106]. Compounds **48** (IC_50_ 7.6, 1.8, and 5.6 μM), **49** (IC_50_ 13, 1.3, and 2.5 μM), and **50** (IC_50_ 1.0, 0.28, and 0.52 μM) exhibited low micromolar activity against A549, CH1, and SW480, respectively, with **50** being the most potent [106]. Furthermore, a 5-nitro derivative (**51**) of these analogs exhibited a significant anticancer activity against HCT-116 and MDA-MB-231 cell lines, with IC_50_ values of 0.302 and 0.738 μM, respectively [107]. This compound also significantly inhibited cell invasion and induced apoptosis in MDA-MB-231 cells, while blocking cell metastasis in a zebrafish human tumor xenograft model without any apparent toxicity [107]. Further assays indicated that compound **51** acts as a dual inhibitor of GSK-3β (IC_50_ 5.46 nM) and Aurora A (IC_50_ 0.22 μM) [107].

2-(*N*-methylpiperazin-1-yl)ethyl and 2-(*N*-methylpiperazinium)ethyl hydrochloride derivatives of these analogs, also exhibited some anticancer activity against LXFL and MCF7 cells in vitro, with compounds **52** (IC_50_ of 0.86 and 0.80 μM) and **53** (IC_50_ of 1.1 and 0.85 μM) displaying the highest potency [72]. Notably, compound **53** exhibited greater water solubility than its counterpart (**52**), and also exhibited selective inhibitory activity on IGF-1R, with an IC_50_ value of 169 nM [72]. This compound also induced apoptosis and caused cell cycle arrest, primarily at the G2/M phase in MCF-7 cells [72]. 

Jeong et al. [108,109,110] further demonstrated that these piperazinium hydrochloride derivatives of indirubin-3′-oxime were potent inhibitors of FLT3 kinase and MV4-11 cells, with compounds **53** and **54** exhibiting the highest potency. Docking studies suggested that compound **53** bound tightly to the ATP binding pocket in FLT3, surrounded by residues Leu616, Leu818, Cys828, and Tyr693, and forming approximately five hydrogen bonds with these residues [108,109]. While the indirubin core formed two significant hydrogen bonds with the backbone Cys694 residue, the tertiary amine of the piperazine moiety formed a hydrogen bond with Asn816, and the terminal amino group established two hydrogen bonds with both Asn816 and Asp829 [108,109]. Moreover, compound **53** effectively inhibited cell growth in vivo in an MV4-11 xenograft model following a daily oral administration of 20 mg/kg, without inducing significant hERG-related cardiotoxicity [108,109,110].

Conversely, compound **52** inhibited the growth of TT thyroid carcinoma cells by suppressing cell proliferation and inducing apoptosis [110]. This compound was also found to suppress the RET signaling pathway via a downregulation of the phosphorylation of RET kinase as well as the downstream Shc and ERK1/2 [108,109].

Dicationic piperazinium dihydrochloride derivatives of indirubin-3′-oxime have also been extensively studied for their anticancer activities. In fact, compound **55** demonstrated activity against Abl1 and T315I mutant Abl1 leukemia cells, with IC_50_ values of 0.87 and 9.4 μM, respectively [111]. This compound also exhibited inhibitory effects on both c-Src and Abl kinases, with SAR studies indicating that the presence of a bromine atom at the 6-position and a flexible alkylamino chain on the 3′-oxime functional group enhances cytotoxicity [111]. Conversely, the trifluoromethyl or bromine group at the 7-position in compound **56** seems to be detrimental to its cytotoxic activity [111]. However, this latter compound showed activity against SH-SY5Y, while displaying only marginal inhibitory activity towards CDKs and GSK-3α/β kinases [112]. Furthermore, compound **56** has been shown to significantly suppress the viability of four human melanoma cell lines (A2058, A375, G361, and MeWo) in vitro in a dose-dependent manner [113]. However, the 5-methyl acetate derivative (**57**) exhibited the highest potency among these analogs, displaying an excellent in vitro FLT3 inhibition, with an IC_50_ value of 3 nM, while blocking the phosphorylation of downstream STAT5 [110]. This same compound exhibited highly selective anticancer activity against leukemia MV-4-11 cells, with an IC_50_ value of 1.2 nM, achieved through G2/M phase arrest and apoptosis induction. Additionally, it inhibited c-Met kinase and blocked c-Met phosphorylation, along with downstream signaling pathways (Erk1/2, STAT3, STAT5, and Akt) in a dose-dependent manner [114]. In an in vivo study, this compound demonstrated a significant inhibitory activity in MV4-11 tumor xenograft model, without causing alterations in the mice’s body weight [110]. Further studies by Vougogiannopoulou et al. [115] indicated that compound **58**, the 6-bromo derivative, exerted strong activity against neuroblastoma SH-SY5Y cells, while also significantly inhibiting GSK-3 activity, with an IC_50_ value of 1.3 nM.

Several indirubin C-3′-oxime ethers bearing a 2-(pyrrolidin-1-yl)ethyl substituent at the oxime functional group have also been shown to strongly inhibit the growth of a series of leukemia cells, while simultaneously blocking the activity of both c-Src and Abl kinases [111]. Among these compounds, **59** and **60** exhibited potent cytotoxicity against wild-type KCL-22 cells and T315I mutant KCL-22 cells in the sub-micromolar range [111]. Additionally, compounds **61** and **62** exhibited significant growth inhibition in SH-SY5Y, HCT116, F1, and Huh7 cell lines [112]. These compounds only displayed marginal inhibitory activity towards CDKs and GSK-3α/β kinases, suggesting that, unlike many indirubin derivatives, their anticancer activities might not be primarily mediated through a direct effect on CDK1, CDK5, and GSK-3α/β kinases [112]. 

Ultimately, Van et al. synthesized a number indirubin-3′-oxime ethers by attaching polyhydric alcohol chains with bulky 1,3,4-thiadiazole substituents to the hydroxyl of the oxime functional group [116]. Among these compounds, **63** and **64** exhibited only a moderate activity, albeit comparable to that of the parent indirubin-3′-oxime [116]. Further SAR studies indicated that the introduction of bulky substituents to the 3′-oxime group mostly resulted in a decrease in the anticancer activity of the resulting molecule [116]. The structure of all these indirubin-3′-oxime ether derivatives are summarized in Figure 4.

While investigating the effect of different substituents at *N*^1^ of the bisindole core structure on the anticancer activity of the resulting indirubin-3′-oxime derivatives, Anh et al. [117] prepared a series of indirubin-based conjugates by attaching a range of hydroxyamides at that position, as illustrated in Figure 5. The inhibitory potential of the resulting molecules on the proliferation of cancer cell lines was investigated, with many of these conjugates exhibiting cytotoxic activity several magnitudes higher than that of the parent indirubin-3′-oximes [117]. Furthermore, compounds **65**–**67** significantly inhibited the proliferation of SW620, PC3, and NCI-H23 cells, with IC_50_ values in the sub-micromolar range. These compounds also displayed strong histone deacetylases (HDACs) inhibitory activity, with IC_50_ values as low as 22 nM. The inhibitory activity of **65** towards HDAC6 (IC_50_ 0.007 μM) was 29-fold higher compared to its effect towards the HDAC2 isoform (IC_50_ 0.205 μM) [117]. Docking studies revealed that **65** positioned into the HDAC6 pocket with the lowest energy and formed three hydrogen bonds with key residues, namely Tyr782, His610, and Asp649. However, it did not display any such hydrogen bond-type interaction within the HDAC2 pocket [117].

The activity of a series of indirubin-3′-oxime conjugates, prepared by attaching different chalcone units at the *N*^1^-position of the core indirubin skeleton via 1,3,4-thiadiazole click chemistry, was also evaluated against four human cancer cell lines (HepG2, LU-1, SW480, and HL-60), with the human normal kidney cell line (HEK-293) used as a control [118]. Among them, compounds **68**–**70** displayed significant growth inhibitory properties on all cell lines, with the most potent compound **68** exhibiting IC_50_ values of 2.01, 1.30, 2.54, and 0.98 μM against HepG2, LU-1, SW480, and HL-60 cell lines, respectively [118]. Docking studies indicated that compound **68** forms three hydrogen bonds towards GSK-3β via two residues, namely Val135 and Thr138 [118]. Further SAR studies indicated that bulky and rigid substituents are not suitable for the 5′-position, as they led to reduced cytotoxicity and poor water solubility. In the NCI60 cell line panel, these compounds generally inhibited the growth of malignant tumor cells at low micromolar concentrations [118].

Other types of bisindole derivatives bearing an oxime moiety, not directly related to indirubin, have also undergone investigation for their anticancer activities. In fact, the antiproliferative activities of a series of 1′H,3H-[2,3′-biindol]-3-ones, designed and synthesized by Qu et al. [119], were explored for their inhibitory effects on seven human cancer cell lines, using HUVEC normal cells as a control. Initial structure–activity relationship (SAR) studies indicated that the presence of an oxime group at position 2 was crucial for the cytotoxic effects. Furthermore, the introduction of a polar substituent at C-5 and C-5′ significantly enhanced the anticancer activities of the resulting compounds [119]. Among these compounds, **71** exhibited the highest potency, with IC_50_ values of 6.09, 7.66, 9.92, and 4.31 μM against A529, MGC-803, HepG2, and T24 cell lines, respectively [119]. 

Subsequent investigations revealed that this compound induced apoptosis in T24 cells by elevating intracellular reactive oxygen species (ROS) levels and altering the balance of anti- and pro-apoptotic proteins, leading to mitochondrial dysfunction and activation of caspase-9 and caspase-3 [119]. Furthermore, both cell cycle analysis and Western blotting indicated that compound **71** effectively arrests the growth of T24 cells in the G1 stage, possibly influencing cell cycle regulatory proteins, particularly cyclin D1, while also causing a significant increase in the activity of p53, p21, and p16 [119].

While assessing a series of bis(indolyl)methane oximes against several tumoral cell lines, namely HepG2 (hepatocellular carcinoma), MDA-MB-468 (human breast carcinoma), RAW 264.7 (murine leukemic monocyte macrophages), THP1 (human acute monocytic leukemia), U937 (human leukemic monocytic lymphoma), and EL4 cells (murine T-lymphoma), Grosso et al. [120] demonstrated that compounds **72** and **73** exhibited potent cytotoxic effects across all tested cells. The IC_50_ values varied between 1.62 (THP1) and 23.9 μM (RAW) for **72**, and between 10.7 (MDA) and 34.1 μM (U937) for **73**. Notably, compound **72** displayed pronounced activity against nonadherent cell lines, with IC_50_ values ranging from 1.62 in THP1 to 1.65 μM in EL4 [120]. In contrast, **74** exhibited a considerably lower cytotoxicity in tumoral cell lines compared to the other two compounds, with IC_50_ values ranging from 35.7 (HepG2) to 124 μM (THP1), without displaying any selectivity. The difference in activity between the *N*-unsubstituted **72** and the *N*-methyl substituted derivative **74** was quite significant, underscoring the potential importance of a free amine interaction within the binding pocket through hydrogen bonds [120]. Conversely, the notably lower IC_50_ values observed for **72** in nonadherent cell lines in comparison to those obtained for **73** demonstrated that the presence of a bromine substituent contributes to higher cytotoxic activity. Unfortunately, these compounds exhibited only a marginal selectivity, with only about one to two-fold difference compared to their activity on nontumoral cells [120].

Dandu et al. [121] prepared and screened a series of dihydroindazolo[5,4-*a*]pyrrolo[3,4-c]carbazole oximes against recombinant human VEGF-R2 and TIE-2 receptor tyrosine kinases. Compounds **75**, **76**, and **77** exhibited activity against both TIE-2 (with IC_50_ values of 30, 25, and 26 nM, respectively) and VEGF-R2 (with IC_50_ values of 7, 4, and 4 nM, respectively), while also demonstrating favorable pharmacokinetic properties in rats [121].

Homology modeling and docking experiments revealed that **75** binds effectively within the ATP pocket of the TIE-2 model, with the lactam NH/CO moiety forming a bidentate donor/acceptor interaction with Glu903/Ala905 at the hinge region, with the anti-oxime orientation serving as an acceptor for the Asp982 backbone amide [121]. The hydrophobic cavity that accommodates the *O*-alkyl oxime is defined by Leu888, Leu976, Phe983, Gly984, Leu985, and Ile902. Indeed, a single-crystal structure solved for **75**, confirmed both the *trans*-oxime orientation and the regiochemistry of the lactam carbonyl, as well as the orientation of the indazole *N*^2^ -methyl group [121].

A number of indeno[1,2-*b*]quinoxaline derivatives were also synthesized and evaluated for their antiproliferative effects. Among them, compounds **78** (IC_50_ 0.87, 5.78, 0.82, and 0.64 μM), **79** (IC_50_ 0.67, 6.88, 0.89, and 0.83 μM), **80** (IC_50_ 0.45, 5.69, 0.68, and 0.82 μM), **81** (IC_50_ 0.90, 0.81, 0.78, and 0.75 μM), **82** (IC_50_ 0.68, 0.92, 0.78, and 0.32 μM), and **83** (IC_50_ 0.66, 0.83, 0.91, and 0.63 μM) demonstrated the ability to inhibit the growth of MDA-MB231, H1299, PC-3, and Huh-7 cancer cell lines, respectively [122]. Compound **78** was found to be inactive against the growth of the normal human fetal lung fibroblast cell line (MRC-5), with an IC_50_ value of 31.51 μM. This study also demonstrated that compounds **78**–**80** (Figure 6) exhibited comparable inhibitory activities against topoisomerase I and topoisomerase II [122]. Mechanistic studies indicated that compound **78** induced cell cycle arrest at the S phase through the activation of caspase-3, -7, an increase in the protein expression of Bad and Bax, and a decrease in the expression of Bcl-2 and PARP, ultimately resulting in cell death [122]. Furthermore, compound **78** attenuated the levels of phosphorylated Src, Akt-1, and Akt-2 proteins, without affecting the total protein expression of Akt. Additional studies involving human hepatocellular carcinoma cells in a zebrafish xenograft assay confirmed the antitumor effect of **78** in vivo [122].A series of diarylmethyloxime derivatives containing 5-indolyl moieties were also found to be potent inhibitors of tubulin polymerization [123]. Their inhibitory activity was assessed against several cancer cell lines, including HeLa human cervix epithelioid carcinoma, A-549 human lung carcinoma, HL-60 human leukemic, and HT-29 human colon adenocarcinoma. Among these compounds, **84** exhibited the highest activity against HeLa, HL-60, and HT-29 cell lines, with IC_50_ values of 0.36, 0.33, and 0.11 μM, respectively. However, this compound showed no activity against the A-549 cell line [123].

A number of indoles psammaplin A derivatives were synthesized by replacing the *o*-bromophenol unit with an indole ring. Biochemical and cellular characterizations conducted on U937 and MCF-7 cells confirmed that many of these analogs exhibited more potent activities compared to the original natural product [124]. Furthermore, in addition to the well-documented dual HDAC and DNMT epigenetic inhibitory profile of the parent compound, some analogs, notably compound **85**, also demonstrated inhibition of NAD^+^-dependent SIRT deacetylase enzymes. The structure–activity relationship (SAR) study provided insights into the mechanism of action underlying these multiple epigenetic ligands, and sets the stage for further structural exploration aimed at optimizing their pharmacological profiles [124]. In fact, enzymatic inhibition studies of HDAC1 and HDAC4 indicated that the C5-Br moiety could be substituted with other halogens (F, Cl, I) or oxygen-containing substituents (OMe, OBn), as these analogs displayed approximately equal potency. However, the absence of a substituent was found to be detrimental to the inhibitory activity [124]. Pharmacokinetic analyses revealed that compound **85** functions as a prodrug that rapidly transforms into a glutathione conjugate. The anticancer effects mediated by **85** seemed to involve the activation of distinct apoptosis pathways in cancer cells due to synergism between its inhibitory activities [124]. Notably, compound **85** demonstrated good tolerability in experimental mouse models, with a maximal tolerable dose higher than that of well-known HDAC inhibitors [124].

### 2.2. Anticancer Activity of Small Nitrogen-Containing Heterocyclic-Based Oximes

Small heterocyclic-based oximes bearing a triazole, pyrazole, imidazole, pyrimidine, xanthine, or quinazoline core structure have also been widely investigated for their anticancer properties, with some illustrative examples provided in Figure 7 and Figure 8.

A series of tubulin polymerization inhibitor hybrid molecules, prepared by replacing the ‘ethylene-bridge’ of Combretastatin A-4 with isooxazoline or triazole heterocyclic pharmacophores, was evaluated for their cytotoxic properties [125]. Many of these compounds exhibited potent antiproliferative activity comparable to that of Colchicine against a breast cancer cell line (MDA-MB-231) and a lung cancer cell line (A-549). Among them, compounds **86** (IC_50_ 0.50 and 0.42 mM), **87** (IC_50_ 0.70 and 0.60 mM), **88** (IC_50_ 0.80 and 0.567 mM), and **89** (IC_50_ 0.80 and 0.76 mM) displayed the best potency against MDA-MB-231 and A-549 cancer cell lines, respectively [125]. These compounds also exhibited significant tubulin depolymerization effects, as indicated by the Western blot analysis and confocal staining assay, with compound **89** also strongly inhibiting tubulin polymerization in vitro [125]. Docking and molecular dynamics studies confirmed the binding interaction patterns of these molecules at the Colchicine binding site of tubulin and convincingly correlated the observed experimental results with the different structural variations. The docking scores and molecular dynamic simulations also correlated with the cytotoxic activity displayed against A-549 and MDA-MB-231 cancer cell lines [125].

A series of 5-(4-pyridyl)-1,2,4-triazole hybrids bearing acetophenone-oxime moieties rationally designed and synthesized were evaluated for their properties as epidermal growth factor receptor (EGFR) kinase inhibitor [126]. Some of these compounds moderately inhibited the growth of the NCI-60 human cancer cell lines when screened in vitro. The most potent oxime analog (**90**) was further evaluated for its EGFR enzyme inhibition activity. This compound exhibited an IC_50_ value of 0.18 µM, in comparison with Gefitinib (IC_50_ 0.06 µM) used as a reference. Docking and molecular dynamic simulations indicated that this compound tightly binds to the EGFR tyrosine kinase binding site [126].

Another series of 1,2,4-triazole/pyrazole hybrids linked to oxime moieties demonstrated potent cytotoxic activities against A-549, MCF-7, HCT-116, and PC-3 cancer cell lines [127]. Among them, the sulphamoyl derivatives with an internal oxime, **91** and **92**, were the most potent derivatives against all tested cell lines, especially PC-3 (IC_50_ = 1.48 and 0.33 µM, respectively), showing roughly 12- and 39-fold selectivity towards PC-3 over F180 fibroblasts, respectively [127]. Mechanistic investigations of **91** and **92** revealed that both compounds arrested the cell cycle at the G2/M phase, upregulated Bax expression, and downregulated the expression of the Bcl-2 gene. These two compounds also proved to be good inhibitors of p38MAPK (IC_50_ 0.65 µM for **91** and 0.58 µM for **92**) and VEGFR-2 (IC_50_ 0.39 µM for **91** and 0.54 µM for **92**), in comparison with PC-3 control cells [127].

Abdel-Aziz et al. [128,129] also revealed that a group of nitric oxide-donating 1,2,4-triazole-oxime hybrids were able to moderately inhibit the growth of several cancer cell lines. Compound **93** achieved cell growth inhibition against both leukemia HL-60(TB) and renal cancer UO-31, while **94** moderately inhibited the growth of non-small cell lung cancer HOP-92 and renal cancer A498 cell lines, and weakly impeded the proliferation of both leukemia K-562 and renal cancer UO-31 cell lines. Additionally, **95** was active against leukemia MOLT-4, RPMI-8226, SR, and renal cancer A498 cell lines, respectively [128,129,130]. As for compound **96**, it moderately inhibited the growth of leukemia HL-60(TB) and RPMI-8226 cell lines, while **97** showed moderate potency against leukemia RPMI-8226, SR, renal cancer CAKI-1, and UO-31 cell lines [128,129,130]. This study demonstrated that analogs with *N*-allyl or *N*-ethyl-substituted triazoles displayed a weak antiproliferative activity, while those with *N*-phenyl-substituted triazoles revealed moderate to weak activity against most of the tested cell lines. Additionally, the unsubstituted 1,2-diphenyl triazole (**98**) displayed the strongest activity among the tested compounds against renal cancer A498 cell lines [128,129,130].

A number of pyrazole oxime ether derivatives have also been studied for their growth-suppressing properties against human solid tumor cell lines. In fact, compounds **99** (IC_50_ 0.12, 0.21, 13.21, 0.04, and 0.02 µg/mL), **100** (IC_50_ 0.10, 0.28, 18.78, 0.02, and 0.01 µg/mL), and **101** (IC_50_ 0.13, 0.92, 4.91, 0.28, and 0.26 µg/mL), all bearing a phenyl residue at position 3 of the core pyrazole, show relatively potent activity against A549, SKOV-3, SKMEL-2, XF 498, and HCT15 cell lines, respectively [131].

Xiong et al. [132] also prepared a series of bis-pyrazole oximes by conjugating thiazolyl-substituted pyrazoles with pyrazoxime, and screened them for their antiproliferative activity against four cancer cell lines in an MTT assay. Compound **102** was the most active across all four cancer cell lines in vitro, with potencies superior to that of 5-fluorouracil (5-FU). Furthermore, this compound selectively promoted intracellular ROS accumulation in HCT116, believed to be a key component of its mechanism of action [132]. This compound also dose-dependently induced cancer cell apoptosis through the regulation of apoptotic protein expression and DNA damage [132].

To further the SAR study for this family of compounds, the same research group designed and synthesized other pyrazole derivatives and investigated their antitumor effects both in vitro and in vivo [47,133]. Several of these compounds displayed good antiproliferative activity, with IC_50_ values in the low-micromolar range against three human cancer cell lines in vitro. The most potent compound (**103**) selectively inhibited the proliferation of SMMC-7721 (IC_50_ 0.76 μM), SGC7901 (IC_50_ 2.01 μM), and HCT116 (IC_50_ 1.26 μM), respectively, without affecting the proliferation of non-tumor LO2 cells [133]. Compound **104** was a close second and even more active against HCT116 (IC_50_ 1.18 μM) than **103**. Additionally, **103** exhibited significant induction of cell apoptosis through marked cleavage of both PARP and caspase-3, with its effects on cell growth inhibition and apoptosis believed to be related to DNA damage and activation of the p53 signaling pathway [133]. This compound also displayed low acute cytotoxicity and significant growth inhibition of cancer cells in vivo [133].

A computer-aided design led to the synthesis of a series of nitroimidazole-oxime derivatives as potential PLK1-PBD inhibitors. Among them, **105** (IC_50_ 0.73, 0.76, and 0.76 µg/mL), **106** (IC_50_ 0.82, 0.71, and 0.79 µg/mL), **107** (IC_50_ 0.03, 0.92, and 0.89 µg/mL), **108** (IC_50_ 0.01, 0.73, and 0.78 µg/mL), **109** (IC_50_ 0.02, 0.81, and 0.81 µg/mL), and **110** (IC_50_ 0.05, 0.75, and 0.72 µg/mL) exhibited potent activity against MGC-803, HepG-2, and MCF-7 tumor cell lines, respectively [134]. Compound **108** exhibited a better biological profile against PLK1-PBD and induced apoptosis in the MGC-803 cell line in a dose-dependent manner. Docking studies indicated that this latter compound positions well into the active center of the polo-box domain of polo-like kinase 1, with the binding model showing that **108** is perfectly bound to the active site PBD domain of PLK1 through three hydrogen bonds with Lys540 and Arg557, one π-cation interaction with Lys540, and one π-π bond with Trp414 [134]. The nitroimidazole group has a good affinity with the Lys540-His538 pincer and clinches in the EBR region, while the 6-methoxypyridine group fits perfectly in the HM region. Furthermore, other weak interactions, such as van der Waals and carbon–hydrogen bonds, also contribute to the binding affinity of **108** with PLK1 [134].

Among a series of alkylthio/sulfinyl-8H-thieno[2,3-b]pyrrolizin-8-oximino derivatives synthesized for their antitumor activities by Guo et al. [135], several were evaluated in vitro against Bel-7402, HT-1080, SGC-7901, and A549 cell lines. Among these compounds **111** displayed a superior antitumor activity to that of Cisplatin [135].

Some pyrimidine-5-carbaldehyde oximes have also been explored for their anticancer activities. In fact, while screening a series of 4-aminopyrimidine-5-carboxaldehyde oxime scaffolds bearing a 4-fluoro-2-methylindol-5-yloxy group at the 6-position and alkyl groups at the oxime sidechains, against a panel of 100 kinases, Huang et al. discovered that many of these analogs possess good potency against the VEGFR family of kinases [136]. Compounds **112** (IC_50_ 0.042 μM) and **113** (IC_50_ 0.024 μM) were highly selective for VEGFR-2 compared to other VEGFR family members (VEGFR-1 and VEGFR-3). These compounds also exhibited only marginal inhibition effects on other angiogenesis-related kinases, including FGFR, PDGFR, and Tie2, while demonstrating potent antiproliferative activity, as illustrated with **112** (IC_50_ 0.032, 0.036, and 0.027 μM) and **113** (IC_50_ 0.31, 0.39, and 0.23 μM), against HeLa, HCT116, and A375 cancer cells, respectively [136]. They cause cell accumulation at the G2/M phase of the cell cycle by preventing cells from entering mitosis [136].

Xu et al. also revealed that 4-amino-6-arylaminopyrimidine-5-carbaldehyde oxime derivatives are potent inhibitors of ErbB-2/EGFR kinase and display antiproliferative effects on ErbB-2 overexpressing SK-BR-3 and BT474 cell lines [137]. Among these compounds, those with a 1-(3-fluorobenzyl)indazol-5-amino group as the C-6 side chain produced optimal dual EGFR and ErbB-2 kinase inhibition, with compound **114** being the most potent against both EGFR and ErbB-2, with IC_50_ values of 8 and 12 nM, respectively [137]. It should be mentioned that a methyl substitution of the C-4 amino group (**115**) abrogated the activity of this family of compounds against both EGFR and ErbB-2. Such a pronounced drop in activity against these enzymes suggests that the NH_2_ group at C-4 is necessary for binding, or that the methyl group prevents the molecule from adopting a proper conformation detrimental to these enzymes’ activities [137]. This observation is supported by the elucidation of the X-ray crystallographic structure of the complex of **116** with EGFR, where the NH_2_ group forms key hydrogen-bonding interactions with the backbone C=O of Met 793, and the oxime sidechain is oriented toward the solvent front [137].

Similarly, 4-amino-6-piperazin-1-yl-pyrimidine-5-carbaldehyde oximes have also been shown to display potent inhibitory effects against FLT3 tyrosine kinase while exhibiting strong antiproliferative activity against both FLT3 ITD-mutated human leukemic and wild-type FLT3 BaF3-expressed cell lines [138]. Compounds **117** and **118** were among the most potent, with IC_50_ values of 1.0 and 3.0 nM against FLT3 tyrosine kinase, respectively [138]. Compound **119**, however, was the most potent against MV4-11 and FLT3 BaF3-expressed cell lines, with IC_50_ values of 78 and 8.0 nM, respectively [138].

Considering the synergistic collaboration between c-Met/HGF and VEGFR-2/VEGF, Qiang et al. synthesized a series of 4-aminopyrimidine-5-carbaldehyde oxime analogs as potential dual inhibitors of both HGF/c-Met and VEGF/VEGFR signaling, aiming to develop broad-spectrum anticancer agents [139]. Compounds **120** (IC_50_ 0.21 and 0.17 µM), **121** (IC_50_ 0.28 and 0.56 µM), **122** (IC_50_ 0.38 and 0.12 µM), and **123** (IC_50_ 0.42 and 0.22 µM) were among the most potent against both enzymes, respectively. In vitro cell proliferation assays revealed that **120** also possesses potent inhibitory activity [139].

Penta-1,4-diene-3-one derivatives bearing a quinazoline and oxime ether moieties have also been shown to exhibit strong inhibitory effects against hepatoma SMMC-7721 cells [140,141], with **124** (IC_50_ 0.64 μM) and **125** (IC_50_ 0.63 μM) displaying better activity than that of gemcitabine (IC_50_ 1.40 μM) [141]. Further mechanistic studies have indicated that many of these compounds effectively control the migration and inhibit the proliferation of SMMC-7721 cells by impeding DNA replication [141]. Western blot results showed that compounds **124** and **125** induced irreversible apoptosis of SMMC-7721 cells by regulating the expression levels of apoptosis-related proteins [141].

A series of 1,3,8-trisubstituted or 1,8-disubstituted xanthine derivatives were tested in a cell viability assay using the human mammary gland epithelial cell line (MCF-10A). All the compounds exhibited no cytotoxic effects, with more than 90% cell viability at a concentration of 50 μM [142]. Compounds bearing an oxime moiety showed superior antiproliferative activity compared to their non-oxime counterparts against pancreatic (Panc-1), breast (MCF-7), colon (HT-29), and lung (A-549) cancer cell lines. Among them, compounds **126** (IC_50_ 0.91, 0.80, 1.70, and 1.50 μM), **127** (IC_50_ 1.30, 1.00, 1.20, and 1.50 μM), **128** (IC_50_ 1.70, 1.30, 1.70, and 1.80 μM), **129** (IC_50_ 1.91, 1.80, 1.70, and 1.50 μM), and **130** (IC_50_ 1.70, 1.40, 1.80, and 1.89 μM) were the most potent against Panc-1, MCF-7, HT-29, and A-549, respectively [142]. These compounds also increased the level of active caspase-3 in Panc-1 cell lines, with **126**, **129**, and **130** being the most potent caspase-3 inducers [142]. Compounds **129** and **130** also increased the levels of caspase-8 and -9, activating both intrinsic and extrinsic apoptotic pathways, with a more noticeable effect on the intrinsic pathway. They also induced Bax and cytochrome C while downregulating Bcl-2 protein levels [142]. Treatment of Panc-1 human pancreatic cancer cells with compound **130** showed cell cycle arrest at the Pre-G1 and G2/M phases, while compounds **126**, **128**, and **130** also inhibited EGFR, with IC_50_ values of 0.32, 0.45, and 0.82 μM, respectively [142].

### 2.3. Anticancer Activity of Quinone-Based Oximes and Oxime Ethers

Quinones are oxidized derivatives of phenolic compounds, with hydroxyl or ether groups increasing the nucleophilicity of the ring and contributing to the large redox potential needed to break aromaticity [143]. Depending on the specific quinone and the site of reduction, they can either re-aromatize or break the conjugation. Consequently, their biological activity is often associated with their redox cycling properties [143,144]. For instance, plastoquinone is well-known for serving as a redox relay in the photosynthetic process, while pyrroloquinoline quinone functions as a biological redox cofactor [145]. Ubiquinones, as implied by their name, are believed to be ubiquitous in the respiratory systems of living creatures [145,146]. Additionally, the auto-oxidation of the neurotransmitter dopamine and its precursor L-Dopa has been shown to generate the comparatively stable dopamine quinone, which inhibits the functioning of the dopamine transporter (DAT) and the tyrosine hydroxylase (TH) enzyme, ultimately leading to low mitochondrial ATP production [147]. This family of compounds is also known to display a wide range of biological and pharmaceutical properties, especially since they constitute a major class of anticancer cytotoxins [144,146]. Consequently, numerous quinone-based oxime derivatives have been investigated for their anticancer activities.

A number of mono-oximes obtained by the oximation of *para*-benzoquinones were investigated for their cytotoxicity on three tumor cell lines, namely HL60, NCI-H292, and MCF-7. Among them, 5-isopropyl-2-methyl-[1,4]-benzoquinone oxime tosylate (**131**, IC_50_ 0.3, 3.6, and 1.6 µg/mL), 5-isopropyl-5-methyl-[1,4]-benzoquinone oxime tosylate (**132**, IC_50_ 0.3, 3.1, and 1.7 µg/mL), and 2,5-dimethyl-[1,4]-benzoquinone oxime tosylate (**133**, IC_50_ 0.3, 5.9, and 1.6 µg/mL) showed significant activity against these cell lines, respectively [148].

Alkannin and shikonin are naturally occurring naphthoquinones primarily found in plants, and are key constituents of the traditional Chinese medicine known as *Zicao*, which is derived from the roots of *Lithospermum erythrorhizon* [149]. These two compounds constitute a pair of enantiomers, differing only by the stereochemistry at carbon 1′ of the prenylated side chain attached at position 2 on the core skeleton (see Figure 8). Alkannin (the *S*-isomer) and shikonin (the *R*-isomer) were initially isolated from the roots of *Alkanna tinctoria* and *Lithospermum erythrorhizon*, respectively [149,150]. This family of compounds has garnered significant interest due to their intriguing biological activities [150,151]. A large number of alkannin and shikonin derivatives have been either isolated from nature or synthesized and evaluated for their tumor-inhibitory potential [150,151].

For example, a set of alkannin and shikonin oxime derivatives were synthesized and their cytotoxicity investigated against three types of tumor cells (DU145, MCF-7, and K562) as well as a normal cell line (HSF) [152]. As a general trend, alkannin derivatives appear to be more active than their shikonin counterparts bearing the same substituents, with many of the synthesized compounds exhibiting better activity and selectivity towards K562 cells, with no toxicity observed in normal cells [152]. Compound **134** demonstrated the highest potency across all cell lines for both the *R* and *S* isomers, while **135**–**139** showed high potency against MCF-7 and K562 cell lines for both isomers.

The same research group expanded their design and synthesis efforts to create a variety of other analogs for a more comprehensive structure–activity relationship (SAR) study by introducing diverse groups or sidechains at the 1′ position, as illustrated in Figure 9 [153,154,155,156]. While the 1′-hydroxy and ether derivatives showed relatively low activity [156], the thio- and benzoylthio-analogs demonstrated significantly superior activity against HCT-15, MGC-803, and K562 cell lines [153,155]. Particularly noteworthy is the fact that the cytotoxic activity of these oxime derivatives against drug-resistant cancer cell lines was much stronger than that against drug-susceptible cell lines. Among the benzoylthio-analogs, **140** (IC_50_ 0.98, 1.21, and 0.85 μM), **141** (IC_50_ 0.50, 0.91, and 0.68 μM), **142** (IC_50_ 0.82, 1.25, and 0.79 μM), and **143** (IC_50_ 0.38, 0.79, and 0.46 μM) displayed significant inhibitory activities against HCT-15, MGC-803, and K562, respectively [153,155]. Similarly, **144** (IC_50_ 0.82, 1.35, and 1.00 μM), **145** (IC_50_ 0.54, 1.09, and 0.76 μM), and **146** (IC_50_ 0.92, 1.54, and 0.98 μM) exhibited the highest potency among the thio-analogs. Compound **143** emerged as the most potent molecule overall, with IC_50_ values ranging from 0.29 to 1.33 μM against MDR sublines [153,155]. Further mechanistic studies indicate that **143** has the potential to inhibit colony formation, induce G1 phase arrest, and promote cell apoptosis by increasing the Bax/Bcl-2 ratio in Bel7402/5-FU cells [153,155].

Subsequent studies revealed that the cytotoxicity of these molecules was not linked to reactive oxygen species (ROS) production or bioreductive alkylation. To identify the molecular target(s) of this molecule, a series of biotinylated derivatives was designed and prepared. Many of these derivatives retained their anticancer activity, including compound **147** (IC_50_ 3.67, 4.76, and 1.26 μM) against K562, MCF-7, and HCT-15, respectively, without exhibiting toxicity against the HSF cell line [154]. Furthermore, cell-based investigations have demonstrated that replacing the C_4_ -linker with a longer chain (C_6_ or C_8_) results in increased cytotoxicity [154].

The Wnt/β-catenin signaling pathway plays a crucial role in the growth, regulation, development, and differentiation of normal stem cells [157,158,159]. Constitutive activation of the Wnt/β-catenin signaling pathway is found in many human cancer cell lines, making it an appealing target for anticancer therapy. Cell-based screening of a library of compounds, along with computational and structure-based design strategies, has led to the development of a series of anthracene-9,10-dione dioximes. When evaluated on a panel of 47 cell lines, including colorectal, lung, ovarian, breast, pancreatic, renal, and hematologic cancers, most of these cell lines showed a sensitivity at concentrations lower than 0.2 μM [160]. The most potent compound (**148**) exhibited IC_50_ values of 0.030, 0.09, 0.020, and 0.01 μM against colorectal (HCT-15, HT-29, HCT-116) and adrenal carcinoma (H296R) cell lines, respectively. Compounds **148** (GIC_50_ 0.09 μM), **149** (GIC_50_ 0.03 μM), **150** (GIC_50_ 0.02 μM), **151** (GIC_50_ 0.08 μM), and **152** (GIC_50_ 0.06 μM) also displayed significant growth inhibition (GI) effects against the HT-29 cell line [160]. Most of the cancer cell lines used in this study exhibited high levels of β-catenin expression when treated with these compounds, indicating that these compounds efficiently modulate β-catenin transcriptional activity and β-catenin cellular levels. Fluorescence-activated cell sorting analysis revealed that compound **148** arrests cancer cell lines in the G1 phase of the cell cycle, and induces apoptosis within 48–72 h of continuous treatment [160]. Consequently, these compounds were recognized as potential agents for treating a broad spectrum of epithelial and hematological malignancies by inhibiting oncogenic Wnt/β-catenin pathway signaling. Furthermore, by blocking the disrupted signaling responsible for early carcinogenesis, metastasis, and the maintenance of the malignant phenotype, compound **148** may prove effective at all stages of a treatment regimen, including metastatic disease, prevention, and the reversal of premalignant, hyperproliferative lesions like colonic polyps [160].

Drug conjugates are chemotherapeutic or cytotoxic agents covalently linked to targeting ligands such as antibodies or peptides via a linker. They have recently gained notoriety as an important class of anticancer therapeutics, and while antibody–drug conjugates (ADCs) are well established in cancer treatment, peptide–drug conjugates (PDCs) are emerging as a new approach for targeted drug delivery with better efficacy and limited side effects [161,162,163,164]. In these conjugates, the linker plays a key role in the drug circulation time as well as in the accurate release of the active entity at the target site for an optimum activity. While early technologies relied on random conjugation to either lysine or cysteine residues, often leading to heterogeneous ADCs, the latest approaches enable the synthesis of homogeneous ADCs via a conjugation at specifically engineered cysteines, glycosylated amino acids, or bioorthogonal unnatural amino acids [165,166]. As a result, targeted therapies based on polysaccharide–drug or protein–drug conjugates have gained prominence in recent years due to their superior efficacy and limited side effects [161,162,163,164,165,166].

Doxorubicin is an antibiotic isolated from the *Streptomyces peucetius* bacterium and belongs to the anthracycline group of chemotherapeutic agents [167,168], which includes daunorubicin, idarubicin, and epirubicin. Doxorubicin, which is marketed under the brand name Adriamycin, among others, has been widely used since the 1960s, and has also been implemented in cancer chemotherapy [169]. This compound is believed to interact with DNA, thereby inhibiting macromolecular biosynthesis [168,169,170] and impeding the progression of topoisomerase II [171]. After disrupting the DNA chain during replication, doxorubicin stabilizes the topoisomerase II complex, impeding the release of the DNA double helix and, as a result, halting the replication process [168].

Schuster et al. [172,173,174,175] reported a series of oxime-linked daunorubicin conjugates, prepared by incorporating a variety of unnatural and *D*-amino acids into the peptide sequence (see Figure 10). These conjugates were gonadotropin-releasing hormone (GnRH) peptide derivatives, designed to target the overexpressed GnRH receptors found in many cancers. All of these conjugates exhibited noticeable activity against MCF-7 breast cancer cells, with IC_50_ values ranging from 0.14 to 6.64 μM, as well as against HT-29 colon cancer cells, with IC_50_ values ranging from 3.31 to 18.00 μM [172,173,174,175]. The lead conjugates **153** and **154** demonstrated stability in 90% human serum for at least 24 h, and ligand competition assays with GnRH receptors indicated that both compounds had a high binding affinity for these receptors. However, flow cytometry revealed that the cellular uptake of **154** appeared to be higher than that of **153** when used to treat MCF-7 and HT-29 cells for 6 h [175]. The same research group further demonstrated that conjugates **153** and **154** possessed potent in vivo activity in breast and colorectal cancer-bearing mice using orthotopic models, including 4T1 and MDA-MB-231 breast cancer and HT-29 colorectal cancer in BALB/c and SCID mice, with the results showing a significant reduction in tumor volume and metastasis inhibition, and lower toxicity effects [175].

PDC **155**, which bears a cathepsin B cleavable peptide (GFLG) intercalated at the *N*-terminal of the targeting peptide sequence and two drug molecules conjugated per peptide, was also shown to exhibit a strong cytotoxicity and a high uptake in PANC-1 cells [176]. In vivo studies in PANC-1 tumor-bearing SCID male mice indicated a significant reduction in tumor volume in **155**-treated mice compared to the control group treated with sterile water, with conjugate **155** being able to significantly inhibit tumor growth even at a low dose of 2 mg/kg with minimal side effects [176].

Since integrins were shown to be overexpressed on the surface of some tumor cells, making them a potential target for cancer drug delivery, Feni et al. [177] used oxime to attach cytotoxic daunorubicin to a linker, employing alkyne−azide click chemistry to construct a consolidated integrin-targeting cyclic diketopiperazine (DKP)-RGD peptide and a cell-penetrating peptide (CPP), to boost effective and targeted cellular uptake. Conjugate **156** exhibited good affinity for tumor-associated integrin receptors, αvβ3 and αvβ5, in an in vitro binding assay. Further studies revealed that the cytotoxicity of **156** (EC_50_ range 2.7−11.0 μM) towards U87, HT-29, and MCF-7 cell lines was less pronounced than that of free daunorubicin, with the conjugate showing no selectivity toward cells with higher expression of integrin αvβ3 (U87) [177].

Gonadotropin-releasing hormone III (GnRH-III), a decapeptide isolated from the sea lamprey brain, is known to exhibit lower potency than GnRH in stimulating gonadotropin secretion [172]. It also exhibited some antiproliferative effects on many tumors expressing the GnRH-R protein [172,173,175]. In their study, Marelli et al. [178] actually demonstrated that oxime bond-linked daunorubicin (Dau)-GnRH-III-derived bioconjugates, in which daunorubicin is coupled to the 8Lys in the native form of GnRH-III, and Dau-[4Lys(Ac)]-GnRH-III, in which daunorubicin is attached to the 8Lys of a GnRH-III derivative where 4Ser is replaced by an acetylated lysine, possess potent antitumor activity. Their research, focused on castration-resistant prostate cancer (CRPC) cells, revealed that both Dau-GnRH-III and Dau-[4Lys(Ac)]-GnRH-III were quickly incorporated into DU145 prostate cancer cells, resulting in a significant cytostatic effect [178]. Additionally, these bioconjugates increased the levels of the active form of caspase-3, suggesting the involvement of apoptosis in their antitumor activity. The antiproliferative effect of both Dau-GnRH-III and Dau-[4Lys(Ac)]-GnRH-III was abrogated by a concurrent treatment of the cells with Antide, an antagonist of the GnRH-R [178]. Furthermore, silencing type I GnRH-R resulted in complete inhibition of the antitumor activity of both bioconjugates. These observations suggest that in CRPC cells, daunorubicin-GnRH-III derivative bioconjugates hinder tumor cell proliferation by stimulating the apoptosis process, and exercise their antitumor effect by activating the expression of the type I GnRH-R in these cells [178].

It should be noted that in an earlier study, Orbán et al. [174] also designed and synthesized three oxime bond-linked daunorubicin-GnRH-III bioconjugates and investigated their in vitro antitumor activity on MCF-7 human breast and HT-29 human colon cancer cells. They analyzed the degradation and stability of these bioconjugates in human serum and in rat liver lysosomal homogenate using liquid chromatography in combination with mass spectrometry. It was found that two out of the three prepared bioconjugates remained stable in human serum for at least 24 h [174]. All three bioconjugates, **157** (I_C50_ 2.2 and 14.2 µM), **158** (IC_50_ 3.9 and 19.4 µM), and **159** (IC_50_ 1.8 and 28.6 µM), exhibited an in vitro antitumor effect on both MCF-7 and HT-29 cell lines. However, their antitumor activity was lower than that of free daunorubicin [174]. These results could be explained by a lower cellular uptake of the bioconjugates, the failure to release the free drug at the tumor site, or by the difference in DNA-binding properties of the resulting metabolites. Furthermore, assessing the DNA-binding behavior of the smallest metabolites found in the lysosomal homogenate, and specifically synthesized for that purpose, indicated that the incorporation of a peptide spacer in the structure of oxime bond-linked daunorubicin-GnRH-III bioconjugates was not a prerequisite for their antitumor activity [174].

With respect to polysaccharide conjugates, using a versatile green oximation reaction, a dextran-doxorubicin conjugate (**160**) was synthesized, and the resulting conjugate (see Figure 10) appeared to spontaneously form micelles with a diameter of approximately 100 nm in an aqueous environment [179]. The particles’ size proved suitable for a selective intra-tumoral accumulation, leveraging the enhanced permeability and retention (EPR) effect, and resulting in improved tolerability, boosted in vivo biodistribution, and a bettered antitumor efficacy for these conjugates [179]. Compound (**160**) efficiently delivered doxorubicin into HepG2 cells due to its acid-sensitive properties, enabling an ‘on-demand’ release of doxorubicin in tumor cells in vitro. Furthermore, when administered via intravenous injection, this compound significantly inhibited tumor growth in a H22-xenografted murine model, with the improved effect attributed to its extended blood circulation, enhanced drug accumulation, and improved intracellular release at the tumor site in comparison to free doxorubicin [179].

### 2.4. Anticancer Activity of Flavonoid-Based Oximes and Oxime Ethers

Flavonoids constitute a large group of polyphenolic secondary metabolites, characterized by a C6-C3-C6 arrangement as their basic skeleton. They comprise approximately 12 structural classes, each distinguished by the degree of oxidation of the C ring, among which are flavones, flavonols, flavans, flavanones, flavanols, chalcones, dihydrochalcones, isoflavones, aurones, anthocyanins, anthocyanidins, and catechins [180,181]. These compounds are well known for their wide range of biological and pharmacological activities, with their cytotoxic and anticancer properties well-documented [181,182,183,184]. Consequently, numerous studies have investigated flavonoid-based oximes for their potential antitumor properties.

Microtubules are highly dynamic structures consisting of α- and β-tubulin heterodimers, and are involved in cell movement, intracellular trafficking, and mitosis [185,186]. With regard to cancer, the tubulin family of proteins is regarded as an effective target for tubulin-binding chemotherapeutics, which operate by restraining the dynamics of the mitotic spindle to induce mitotic arrest and cell death [185,186]. Furthermore, changes in microtubule stability, the expression level of different tubulin isotypes, and altered post-translational modifications have been reported in several types of cancer. Emerging evidence has also suggested that tubulins and microtubule-associated proteins might be involved in a number of cellular stress responses, granting a survival advantage to cancer cells [185,186].

Chiang et al. [187] designed and synthesized a series of flavone-based oximes, and evaluated them for their antiproliferative effects on a range of cancer cell lines by monitoring proliferation, cell cycle, apoptosis, caspase levels, and mitochondrial membrane potentials. Among these molecules, **161** (see Figure 11) was the most potent, exhibiting IC_50_ values of 0.45 and 0.43 μM against drug-sensitive NPC-TW01 and HONE-1 cancer cell lines, respectively [187]. The same molecule showed potency against drug-resistant TW01-Taxol-Res and TW01-Met-Res cell lines, with IC_50_ values of 0.57 and 0.66 μM, respectively, while having little effect on the non-cancer human fibroblast Detroit 551 cell line (IC_50_ > 20 μM) [187]. This compound also induced cell accumulation in G2/M, and apoptosis in a time- and concentration-dependent manner. Colchicine competition-binding experiments and computer modeling also indicated that compound **161** induces microtubule disruption by binding to the colchicine-binding site of tubulin, leading to mitochondrial membrane damage and cell apoptosis through the activation of caspase-9/-3, without noticeable activation of caspase-8 [187]. Furthermore, in vivo studies revealed that at doses of 25 and 50 mg·kg^−1^, this molecule displayed good pharmacokinetic properties, and completely inhibited the growth of NPC-TW01 cells in a xenograft nude mouse model [187].

A number of flavanol-based oximes, including *E*-hesperetin oximes (**162** and **163**) have also exhibited noticeable cytotoxicity against human cancer cell line SGC-7901 [188]. A naringenin-based oxime (**164**) was also found to exhibit cytotoxic, genotoxic, and apoptotic effects while inducing and increasing the production of reactive oxygen species in a dose-dependent manner when evaluated against MCF-7, HT-29, and PC-12 cancer cell lines [189]. Strong correlations were observed between cell viability, DNA damage, and apoptosis in all cell lines treated with these compounds, with the oxime derivative outperforming the parent naringenin in all of these categories [189]. Furthermore, several naringenin oxime derivatives, including **165** (IC_50_ 3.32 µg/mL), **166** (IC_50_ 3.63 µg/mL), and **167** (IC_50_ 4.59 µg/mL), exhibited stronger antiproliferative activity compared to naringenin [190]. In fact, the conversion of the inactive 7,4′-di-*O*-propylnaringenin (IC_50_ > 100 µg/mL) into its oxime derivative resulted in an approximately 22-fold increase in potency compared to the parent compound [190].

Another series of naringenin-based oxime ethers (**168**–**171**) demonstrated significant antiproliferative activity against human leukemia (HL-60), gynecological isolated cervical (HeLa, Siha), and breast (MCF-7, MDA-MB-231) cancer cell lines [191]. Additional cell cycle analysis showed that these compounds caused a noticeable increase in the hypodiploid (subG1) phase and induced apoptosis in HeLa and Siha cells, while leading to cell cycle arrest in the G2/M phase in MCF-7 cells. The proapoptotic potential of some selected compounds was further established by the activation of caspase-3 [191]. While evaluating the antioxidant activities of these molecules using xanthine oxidase, DPPH, and ORAC assays, the methyl-substituted oxime ether (**168**) exhibited the strongest activity [191]. In contrast, neither naringenin nor the *Z*-isomer of compound **164** inhibited the growth of any of the tested cancer cell lines at any of the tested concentrations, while the allyl derivative (**170**) and the *E*-isomer of **164** displayed only a marginal growth inhibition activity [191]. The benzyl derivative (**171**) moderately inhibited the growth of MCF-7 cells, a level of activity similar to that of the methyl derivative (**168**) on HL-60 cells [191]. Metodiewa et al. [192] also revealed that *N*,*N*-diethylaminoethyl flavanone oxime ethers (**172**–**175**) induce programmed cell death (apoptosis) in Yoshida Sarcoma cells in vivo, with the administration of these compounds in the promotion phase of the disease leading to both growth inhibition (cell cycle perturbation) and apoptosis [192].

A series of flavanone-triazole hybrids were also synthesized and tested for their cytotoxicity against HCT-15, HeLa, NCI-H522, and HEK-293 (normal) cell lines, with compounds **176**–**179** exhibiting significant cytotoxicity across all the cancer cell lines. Compound **178** was more potent against NCI-H522 cell line, while **176** and **177** appeared to be more toxic against the normal cell line HEK-293 [193].

Some chalcone oxime derivatives have also been evaluated as potential tubulin polymerization inhibitors, and for their cytotoxicity against human cancer cell lines, with many of them demonstrating remarkable inhibitory activity against tubulin polymerization by competing for the colchicine-binding site on tubulin [194]. This activity appeared to be associated with G2/M phase cell cycle arrest as well as favorable antiproliferative effects [194]. In fact, compound **180** was shown to possess potent tubulin inhibitory activity (IC_50_ = 1.6 μM) and potent antiproliferative properties against A549, HeLa, and MCF-7 cell lines, with GI_50_ values of 2.1, 3.5, and 3.6 μM, respectively [194]. Compounds **181** (GI_50_ 2.8, 6.6, and 5.8 μM) and **183** (GI_50_ 3.2, 7.7, and 8.9 μM) also exhibited decent antiproliferative activity against these cancer cell lines [194]. Docking simulations based on the tubulin crystal structure (PDB code: 1SA0) revealed that compound **180** binds tightly to the colchicine domain of tubulin [194].

Furthermore, several chalcone-based oximes containing a ligustrazine moiety were tested for their in vitro antiproliferative inhibitory activity using A-375, MCF-7, A-549, HT-29, and H-460 human cancer cell lines [195]. The effects of these analogs on several cancer markers, including EGFR, BRAFV600E, c-Met, and tubulin polymerization, were also investigated with the objective of identifying their putative target(s).The majority of these compounds exhibited strong anticancer activity, with **183** (IC_50_ 2.43, 0.85, 1.42, 7.83, and 1.83 μM), **184** (IC_50_ 1.47, 0.79, 1.32, 3.80, and 1.63 μM), and **185** (IC_50_ 0.87, 0.28, 1.25, 2.43, and 1.04 μM) being among the most potent against A-375, MCF-7, A-549, HT-29, and H-460 cell lines, respectively, compared to foretinib (IC_50_ = 1.9, 1.18, 1.15, 3.97, and 2.86 μM), used as a positive control [195]. A careful examination of the structure of these molecules reveals that a *Z*-configuration for the oxime functional group is a requirement for the antiproliferative activity, along with a 2-thiophenyl or a 2-phenylthiophenyl moiety, with or without a bromo group at position-5. Molecular analysis indicates that compound **184** is more active than foretinib in inhibiting EGFR, while compound **185** displayed similar activity to that of foretinib in inhibiting BRAFV600E and c-Met [195]. The inhibition of tubulin polymerization by many of these compounds was reminiscent of that of colchicine [195].

### 2.5. Anticancer Activity of Miscellaneous Aromatic- and Heterocyclic-Based Oximes and Oxime Ethers

Other miscellaneous heterocyclic-based oximes and oxime ethers have also been evaluated for their anticancer activity. In fact, while evaluating a series of 4-anilinofuro[2,3-b]quinolines-based oximes in vitro for their anticancer activity against MCF7, NCI-H460, and SF-268 cell lines, Chen et al. [196] revealed that compounds **186** (IC_50_ 0.63 μM, 9.34 μM, and 6.26 μM) and **187** (IC_50_ 0.71 μM, 31.01 μM, and 9.14 μM) possess weak potency against these cancer cell lines, respectively. These compounds also exhibited a high degree of selectivity when compared to their effect on the human lung fibroblast cell line (MRC-5), with IC_50_ values of 42.85 μM and 47.04 μM, respectively. Mechanistic studies indicated that compound **186** may induce apoptosis in cancer cell lines through a mitotic arrest and mitotic catastrophe process [196]. The pharmacokinetic properties of the hydrochloride salt of **186** demonstrated excellent water solubility (1049 μg/mL) as well as high oral bioavailability (57.1%), with a moderate plasma half-life (3.4 h) [196]. Following an oral dose of 20 mg/kg, the hydrochloride salt of **186** exhibited fast oral absorption in mice, with a short Tmax of 0.5 h, and a significant second absorption peak appearing at 1.5 h after dosing. Xenograft studies indicated a significant reduction in tumor size in nude mice treated with this salt, with its evaluation in an orthotopic lung cancer model further revealing that the hydrochloride salt of **186** can be readily absorbed through oral administration and quickly distributed throughout lung tissues, while demonstrating strong inhibition effects on the growth of lung cancer cells [196].

Synthesized derivatives of quinolin-2(1H)-one bearing either an oxime or amide group, were also evaluated for their antiproliferative and antiplatelet activities [197]. Initial assays showed that amide derivatives were either marginally active or completely inactive, while their oxime counterparts displayed noticeable inhibitory activities against platelet aggregation induced by collagen, AA (arachidonic acid), and U46619 (the stable thromboxane A(2) receptor agonist) [197]. Among these compounds, (*Z*)-6-[2-(4-methoxyphenyl)-2-hydroxyiminoethoxy]quinolin-2(1H)-one (**188**) appeared to be the most potent against AA-induced platelet aggregation, with an IC_50_ of 0.58 μM, but exhibited no cell proliferation inhibition effects [197]. Compounds **189** (IC_50_ 0.56 μM), **190** (IC_50_ 0.58 μM), and **191** (IC_50_ 0.54 μM) exhibited potent inhibition activities against U46619-induced aggregation [197].

Curcumin-based oxime analogs have also undergone scrutiny for their potential as anticancer agents [198,199,200,201]. Qin et al. [198,199] reported the antiproliferative activity of a series of oxime and oxime ether analogs of cyclohexanone, piperidin-4-one, and tetralone (see Figure 12), which were tested against various cancer cell lines, including HT-29, PC-3, A-549, Panc-1, MCF-7, H-460, and PaCa-2. Among these analogs, **192** (IC_50_ 0.09, 0.05, 0.08, 0.09, 0.07, 0.08, and 0.10 μM), **193** (IC_50_ 0.03, 0.04, 0.04, 0.05, 0.04, 0.03, and 0.04 μM), and **194** (IC_50_ 0.02, 0.03, 0.02, 0.04, 0.02, 0.04, and 0.02 μM) were the most potent [199]. A mechanistic study revealed that compounds **194** and **195** exhibited potent activity against various targets, including BRAFV600E (with IC_50_ values of 1.3 μM for **194** and 1.5 μM for **195**) and EGFR-TK kinases (with IC_50_ values of 0.04 μM for **194** and 0.05 μM for **195**).

Further structure–activity relationship study led to other potent analogs, including **196** (IC_50_ 0.2, 0.07, 0.4, 0.3, 0.4, 0.4, and 0.4 μM), **197** (IC_50_ 0.09, 0.07, 0.06, 0.06, 0.07, 0.08, and 0.09 μM), **198** (IC_50_ 0.09, 0.09, 0.07, 0.08, 0.09, 0.08, and 0.02 μM), **199** (IC_50_ 0.02, 0.02, 0.09, 0.02, 0.03, 0.04, and 0.02 μM), and **200** (IC_50_ 0.09, 0.08, 0.09, 0.07, 0.09, 0.07, and 0.08 μM) against the same cancer cell lines, with **199** demonstrating potency comparable to that of the positive control Erlotinib [198,199]. While assessing their mechanistic effects on BRAFV600E, EGFR TK kinases, and tubulin polymerization, as well as for their potential to reverse efflux-mediated resistance developed by cancer cells in vitro, compound **198** strongly inhibited the activity of BRAFV600E, with an IC_50_ value of 0.9 μM, while the inhibitory activity of its analog **197** was more pronounced against EGFR TK, with an IC_50_ of 0.07 μM [199]. Some of the analogs, including **196**, **198**, and **199**, exhibited a dual potential as both anticancer agents and MDR reversal agents [199].

Following a trend of strong anticancer activity observed within this family of compounds, the same research group prepared a number of ligustrazine-containing cyclohexanone- and piperidin-4-one-based oxime analogs, and tested them against A-549, PC-3, MCF-7, PaCa-2, and HT-29 cancer cell lines [200]. Among them, **201** (IC_50_ 0.08, 0.07, 0.04, 0.09, and NA μM), **202** (IC_50_ 0.01, 0.009, 0.01, 0.01, and 0.01 μM), **203** (IC_50_ 0.05, 0.04, 0.03, 0.03, and 0.05 μM), **204** (IC_50_ 0.09, 0.07, 0.08, 0.09, and 0.10 μM), **205** (IC_50_ 0.07, 0.05, 0.08, 0.08, and 0.06 μM), and **206** (IC_50_ 0.02, 0.03, 0.04, 0.03, and 0.04 μM) emerged as the most potent across these cell lines [200]. The evaluation of these compounds for their effect on tubulin polymerization, EGFR TK kinases, KAF and BRAFV600E, as well as for their effect on reversing the efflux-based resistance developed by cancer cells, revealed that EGFR was strongly inhibited by **202** and **204**, with analogs bearing 1-isopropyl-piperidin-4-one linker showing the best inhibition against KAF [200]. Compounds **201**–**203** exhibited a dual role as anticancer and as multidrug resistance (MDR) reversal agents. Molecular docking studies indicated that **203** fits well inside the ATP binding pocket of the FAK kinase, with the piperidinone core nestled between Ile428, Gly429 and Glu506 residues within the kinase hinge region [200]. On the other hand, the pyrazine ring bearing three methyl groups generates hydrophobic contacts with Gly429, Glu500, Asp564 and Leu567, located near the gatekeeper residue Met499 and DFG motif of the FAK activation loop. This compound also formed an additional H-bond interaction between the oxime moiety in compound **203** and the Ile428 residue in the hinge region of FAK.

Furthermore, a series of 1-[3-{3-oxopropyl]-4-piperidone oximes, prepared as potential antineoplastic agents, were also evaluated against a panel of leukemic human colon cancer cell lines, including Colo205, HCC-2998, HCT-116, HCT-15, HT29, KM12, and SW-620. Among them, compounds **207** (IC_50_ 1.50, 1.41, 0.22, 0.37, 0.32, 0.23, and 0.20 μM), **208** (IC_50_ 0.35, 0.31, 0.21, 0.27, 0.30, 0.22, and 0.15 μM), **209** (IC_50_ 0.39, 0.36, 0.32, 0.27, 0.32, 0.25, and 0.23 μM), **210** (IC_50_ 0.40, 1.17, 0.49, 0.36, 0.33, 0.29, and 0.20 μM), and **211** (IC_50_ 1.15, 1.58, 0.36, 0.55, 0.35, 0.28, and 0.29 μM) displayed strong cytotoxicity against these cell lines, respectively [201]. As a general trend, quantitative structure–activity relationships revealed that both the cytotoxic potency and selectivity index figures increased as the magnitude of the Hammett sigma values rose. Furthermore, compounds **212** and **213** appeared to lower the mitochondrial membrane potential in CEM cells, while **214** induced transient G2/M accumulation in Ca9-22 cells [201].

The conversion of the 4′-carbonyl of the natural product griseofulvin into an oxime moiety as a strategy to improve its solubility and metabolic stability resulted in analogs **215** (IC_50_ 1.29 and 1.61 μM) and **216** (IC_50_ 10.1 and 14.13 μM), which showed better cytotoxicity than the parent natural product against MDA-MB231 and U2OS cancer cells, respectively, with compound **216** exhibiting an about 130-fold increase in PBS solubility compared to the parent compound [202]. Similarly, a series of oxime ester derivatives of the natural product schizandrin, the major compound isolated from *Schisandra grandiflora*, were evaluated in vitro for their antiproliferative activities against a selected panel of human cancer cell lines (A549, RKO P3, DU145, and HeLa), with normal cells (HEK293) used as a control [203]. Several of these derivatives were found to be more potent than the parent compound, notably, **217** and **218**, which exhibited a strong activity against DU-145 and RKO P3 cell lines, with IC_50_ values of 3.42 µM and 3.35 µM, respectively. Cell cycle analysis indicated that both compounds were capable of inducing apoptosis and cell cycle arrest in the G0/G1 phase [203]. Moreover, the tubulin polymerization assay showed that **217** and **218** can significantly inhibit tubulin assembly, with molecular docking studies and competitive binding assays revealing that these two compounds effectively bind to the colchicine binding site of tubulin [203].

Radicicol is a macrocyclic antifungal antibiotic known to bind to the heat shock protein 90 (Hsp90) chaperone, impeding its function [204,205]. Hsp90 family chaperones are well known to be associated with several signaling molecules, and to play an essential role in signal transduction critical for tumor cell growth [204,205,206].

Due to radicicol’s failure to produce any antitumor activity in vivo in experimental animal models, radicicol 6-oxime and its analogs (Figure 13) have been investigated for their effects on human tumor cell growth, both in vitro and in vivo. As a result, compound **219** demonstrated potent antiproliferative activities against various human tumor cell lines in vitro while inhibiting v-src- and K-ras-activated signaling [207]. Additionally, this compound depleted Hsp90 family chaperone-associated proteins, such as p185erbB2, Raf-1, cyclin-dependent kinase 4, and mutant p53, at the dose needed for the antiproliferative activity. Compound **219** also competed with geldanamycin for binding to Hsp90, in contrast to **220**, the inactive methylether oxime analog of radicicol, which was less potent both in p185erbB2 depletion and Hsp90 binding [207]. More importantly, **219** demonstrated significant inhibition of the growth and proliferation of human breast carcinoma MX-1 cells transplanted into nude mice, administrated at a dose of 100 mg/kg twice daily for five consecutive intravenous injections [207]. The same compound exhibited significant antitumor activity against human breast carcinoma MCF-7, colon carcinoma DLD-1, and vulval carcinoma A431 cell lines in vivo, in an animal model [207]. The depletion of Hsp90-associated signaling molecules (Raf-1 and cyclin-dependent kinase 4) was also confirmed through ex vivo Western blotting analysis using MX-1 xenografts. These findings, in agreement with in vivo antitumor activity, suggests that the antitumor activity of this compound may be mediated, at least partially, by binding to Hsp90 family proteins, leading to the destabilization of Hsp90-associated signaling molecules [207].

The mechanism of action of the *E*-isomer of another oxime derivative of radicicol (**221**) was investigated in a comparative analysis to its *Z*-isomer by examining its inhibitory effects on Hsp90 function and apoptosis induction in erbB2-overexpressing human breast carcinoma KPL-4 cells, in vitro [208]. Direct binding activity to Hsp90 was assessed through Hsp90-binding assays using geldanamycin or radicicol beads. The plasma concentrations of these compounds after intravenous injection in BALB/c mice, and their antitumor activity were studied against KPL-4 cells transplanted into nude mice in an animal model, with the inhibition of Hsp90 function and the induction of apoptosis also evaluated in vivo using tumor specimens from drug-treated animals [208]. It turned out that **221** possesses a strong antiproliferative activity against all breast cancer cell lines tested in vitro, with its activity surpassing that of its *Z*-isomer (**222**) [208]. These results appeared to be in line with the ability of **221** to deplete Hsp90 client proteins and to induce apoptosis in KPL-4 cells in vitro, especially since this compound, but not its *Z*-isomer, showed significant in vivo antitumor activity, followed by the induction of apoptosis in KPL-4 human breast cancer xenografts [208]. While the plasma concentrations of these compounds were similar, only compound **221** (but not **222**) was able to deplete Hsp90-related proteins such as erbB2, Raf-1, and Akt in the tumor specimens retrieved from nude mice [208]. These data indicated that the depletion of Hsp90 client proteins in tumor cells as a result of the inhibition of the Hsp90 function, plays a role in the observed anticancer activity of **221**, and that the stereochemistry of the oxime functional group might be critical for the activity of radicicol oxime derivatives [208].

Two other oxime ethers, **223** (IC_50_ 0.41, 0.44, and 0.58 μM) and **224** (IC_50_ 0.36, 0.54, and 0.57 μM), showed similar potency to that of **219** (IC_50_ 0.24, 0.39, and 0.38 μM) in antiproliferative assays against MX-1, MCF-7, and A431 cancer cell lines, as well as in tyrosine kinase assays, while exhibiting potent activity in vivo using subcutaneous inoculation in animal tumor models [209]. Following the preliminary structure–activity relationships, this research group designed and synthesized radicicol-based probes (**225** and **226**), which could find application in the elucidation of the role and functions of Hsp90 in cellular processes [209].

Furthermore, a series of *O*-carbamoylmethyloxime derivatives of radicicol were also evaluated for their in vitro antiproliferative activities against v-src- and K-ras-transformed cells and for their inhibitory activity against v-src tyrosine kinase [210]. *O*-(piperidinocarbonyl)methyloxime (**227**) demonstrated a superior antiproliferative activity when compared to radicicol and its oxime derivative **219**, with an IC_50_ value of 25 nM for the inhibition of v-src kinase activity [210]. Compound **227** also decreased the Raf-1 protein level in KNRK5.2 cells and demonstrated a potent antitumor activity against MX-1 and A431 xenografts in nude mice [210].

### 2.6. Anticancer Activity of Small Aromatic and Phenolic-Based Oximes and Oxime Ethers

Some small aromatic and phenolic-based oximes have also been evaluated for their anticancer properties, with the structures of all the selected examples provided in Figure 14. While investigating cytotoxicity in a series of thioaryl naphthyl-methanone-based oxime ether analogs towards various cancer cells, 4-(methylthio)phenyl)(naphthalen-1-yl)methanone *O*-2-(diethylamino)ethyl oxime (**228**) exhibited the best safety profile [211]. This compound was able to induce apoptosis, impede migration and invasion, while strongly inhibiting the cancer stem cell population and demonstrating potent tumor regression in mouse MCF-7 xenografts after oral administration [211]. Mechanistic studies indicated that this molecule strongly inhibited EGF-induced proliferation, migration, and tyrosine kinase (TK) signaling in breast cancer cells. However, this compound could not directly inhibit EGFR or other related receptor TKs in a cell-free system [211]. While further investigating the putative target of these compounds upstream of EGFR, it was discovered that the biological effects of **228** could be reversed by the pertussis toxin [191].

Estrogens are known to play diverse roles in various pathophysiological conditions [212,213,214]. A panel of synthetic selective estrogen receptor modulators (SERMs), including tamoxifen and raloxifene, was prepared and used to treat ER-related diseases, including breast cancer and osteoporosis [212,213,215]. Among them, bis(4-hydroxy)benzophenone oxime ether derivatives, including **229** (IC_50_ 75 nM), **230** (IC_50_ 46 nM), and **231** (IC_50_ 77 nM), were shown to possess estrogen receptor (ER) agonist properties associated with noticeable antiproliferative activities arising through an ER-independent mechanism in cancer cells [216]. Furthermore, compound **231** and its analog **232** were shown to upregulate the activity of an estrogen response element (ERE)-containing luciferase reporter construct (ERE-luciferase) in MCF7 cells [217]. Notably, **231** showed higher activity than E2, the endogenous ligand, at concentrations of 1000 nM and 100 nM, but exhibited only a marginal activity at 10 nM [217]. The same compound also exhibited a greater agonistic effect than its counterpart (**232**), inducing a concentration-dependent stimulation of ERE-mediated transcriptional activity with an EC_50_ (median effective concentration) value of 46.3 nM, while inhibiting the androgen receptor-dependent transcriptional activity [217]. It also induced activation-related phosphorylation of ER and enhanced the transcription of ‘growth regulation by estrogen in breast cancer 1′ (GREB1), further supporting its ER-stimulating activity [217]. Additionally, compound **231** exerted antiproliferative effects on various cancer cell lines, including an ER-negative breast cancer cell line, suggesting that it is capable of suppressing the growth of cancer cells independent of its ER-modulating activity [217]. Additionally, a treatment with **231** resulted in a significant apoptotic death of MCF7 and Ishikawa cancer cells, but not in non-cancer human umbilical vein endothelial cells [217].

Targeting tubulin polymerization, a series of 5-(furan-2-yl)-4-(3,4,5-trimethoxyphenyl)-3H-1,2-dithiol-3-one oximes were prepared and investigated for their antiproliferative activities [218]. Among them, **233** (IC_50_ 2.33, 2.72, and 1.96 μM) exhibited potent antiproliferative activity against human gastric adenocarcinoma SGC-7901, human non-small cell lung cancer A549, and human fibrosarcoma HT-1080, respectively [218].

Compound **233** was further assessed against a panel of cancer cell lines, including HeLa (IC_50_ 1.84 μM), MCF-7 (IC_50_ 7.98 μM), A549 (IC_50_ 2.72 μM), HepG-2 (IC_50_ 2.24 μM), and KB (IC_50_ 2.22 μM) [218]. Furthermore, this compound exhibited strong antiproliferative effects against HT-1080 cells in a time- and dose-dependent manner, with the growth inhibitory effect appearing to be related to microtubule depolymerization [218]. Molecular docking studies revealed that **233** interacts and binds efficiently with the colchicine-binding site of tubulin. In addition, a treatment with **233** induced G2/M cell cycle arrest dose-dependently, and subsequently induced cell apoptosis, with the effect associated with both the mitochondrial and death receptor pathways [218]. Western blot studies indicated that upregulation of cyclin B1 and p-cdc2 was related to G2/M arrest [218].

Vascular endothelial growth factor receptors (VEGFRs), a family of receptor protein tyrosine kinases, are known to play a critical role in the regulation of tumor-induced angiogenesis [219]. Consequently, VEGFR inhibitors have been widely used in the treatment of various tumors [219].

Following a hypothesis that a pseudo six-membered ring formed through intramolecular hydrogen bonding could be exploited to mimic planar quinazolines, a series of potential VEGFR-2 inhibitors containing an oxime as a hinge-binding fragment were designed and synthesized [220]. Most of the compounds tested in this study showed potent VEGFR-2 inhibitory activity, especially **234** (IC_50_ 8.7 nM), **235** (IC_50_ 5.3 nM), and **236** (IC_50_ 4.2 nM), which exhibited significant enzymatic inhibitory activity as well as potent antiproliferative activity against cancer cells [220]. Molecular docking suggested that the salicylaldoxime moiety formed two hydrogen bonds with the hinge region [220].

An oxime-bearing derivative of 2-((2,4,5-trifluorobenzyl)oxy)benzaldehyde (**237**) was also shown to suppress leukemic cell growth by significantly increasing reactive oxygen species (ROS) levels and inducing cell death [221]. This compound induced apoptotic cell death, as indicated by nuclear condensation, DNA fragmentation, and annexin V staining. This compound also elevated Bax/Bcl2 levels and caspase9, and caspase3/7 activity, while decreasing mitochondrial membrane potential [221]. ROS production was reduced by *N*-acetyl-*L*-cysteine (a ROS scavenger) and diphenyleneiodonium chloride (a nicotinamide adenine dinucleotide phosphate (NADPH) oxidase inhibitor) after exogenous treatment with **238**, suggesting that this latter compound promotes apoptosis by modulating ROS levels and regulating NADPH oxidase activity [221]. On the other hand, a series of oxime ether derivatives of phenylpropanoids was shown to display in vitro anticancer activities when evaluated against human cancer cell lines, with compounds **238** (IC_50_: 0.443 and 0.225 µM) and **239** (IC_50_: 0.513 and 0.407 µM) exhibiting significant potency against lung cancer (A549) and gastric cancer (SMMG-803) cell lines, respectively [222].

While evaluating structurally modified analogs of the antimitotic natural product Curacin A for their antiproliferative activity on breast (MDA-MB231), prostate (PC3), and ovarian (2008) human tumor cell lines, two oxime analogs, **240** (GI_50_: 0.12, 0.25, and 0.24 µM) and **241** (GI_50_: 0.98, 1.2, and 9.9 µM), displayed good antiproliferative activity against the three cell lines, respectively [223]. Surprisingly, the less lipophilic and structurally simplified oxime (**241**), while displaying slightly weaker growth inhibition properties than the parent oxime analog (**240**), was more potent than Curacin A at inhibiting the assembly of purified tubulin, with the oxime moiety likely acting as a bioisostere of the (*Z*)-alkene group [223].

Some Suberoyl anilide hydroxamic acid (vorinostat) analogs with the aziridin-1-yl oxime moiety, including compounds **242** (IC_50_ 0.60 and 0.90 mM), **243** (IC_50_ 0.30 and 1.0 mM), and **244** (IC_50_ 0.60 and 0.90 mM) were shown to possess high antiproliferative activity against HT1080 and MG22A human cancer cell lines, while displaying only weak histone deacetylase inhibition activity in HeLa cell line extracts [224].

Psammaplin A (**245**) is a marine metabolite previously reported to be a strong inhibitor of two classes of epigenetic enzymes: histone deacetylases and DNA methyltransferases. A library of psammaplin A-based core analogs and moieties was synthesized, and their cytotoxicity was evaluated in A549, MCF7, and W138 cell lines [225]. Many of the compounds, including **246** (IC_50_ 0.16, 0.61, and 1.13 µM) and **247** (IC_50_ 1.59, 1.13, and 7.86 µM), showed significant cytotoxicity in A549, MCF7, and W138 cells, respectively, with the cytotoxicity SAR correlating with HDAC inhibition [225]. Compound **245** (IC_50_ 7.5, 1.27, and 3.44 µM) was demonstrated to function as a natural prodrug, with the reduced form being highly potent against HDAC1 in vitro (IC_50_ 0.9 nM) [225]. Furthermore, this compound exhibited high isoform selectivity, with 360-fold potency for HDAC1 over HDAC6 and more than 1000-fold less potent against HDAC7 and HDAC8 [225]. Moreover, treatment with this compound caused upregulation of histone acetylation, but had only a marginal effect on tubulin functions, with no evidence for **245** acting as a DNA methyltransferase (DNMT) inhibitor [225].

### 2.7. Anticancer Activity of Steroidal Oximes and Oxime Ethers

Steroids are an important class of natural products, characterized by a basic molecular structure typically made up of 17 carbon atoms arranged in a consecutively fused four-ring system [226,227]. Three of these rings are six-membered cyclohexane rings (rings A, B, and C), while one is a five-membered cyclopentane ring (the D ring), as illustrated in Figure 15. Additional carbons are often added to the structure through the incorporation of methyl, prenyl, or isoprenyl groups at different positions, ranging from estrane (18 carbons), androstane (19 carbons), pregnane (21 carbons), all the way to cholestane (27 carbons) [226,227]. They are known to regulate several key processes in various biological systems, and considering their compelling biochemical properties, including their ability to easily penetrate cell membranes and bind to nuclear and membrane receptors, as well as their suitability for structural modifications, steroids represent an extremely attractive motif for designing potential new treatments for a range of diseases. In fact, since their discovery in 1935, steroids have been widely used in the treatment of several conditions, including autoimmune and inflammatory diseases and cancers [228,229]. Because even minor alterations to their core skeleton can result in significant variations in biological activity, a large number of steroidal derivatives with a wide range of therapeutic activities have been reported [43,229]. Exploiting these attributes, the hydrophobic steroid core has been used as a framework for a variety of oxime-bearing derivatives and analogs, which have exhibited remarkable antiproliferative and antitumor properties in many cases [228]. Interestingly, the introduction of this functional group frequently improves the bioactivity when compared to non-oxime analogous compounds. These improvements can be attributed to either the enhancement of the overall biochemical properties of the resulting molecules or to additional interactions with biological targets [51,230]. Furthermore, it has been observed that varying the position of the hydroximino group on the parent skeleton leads to remarkable changes in the relevant activity.

While screening a small library of steroidal oxime ethers with either an androstene or estrane core skeleton on a panel of cancer cell lines, including NCI-H23, NCI-H522, MDA-MB-231, MDA-MB-435, SW-620, COLO 205, LOX, UACC-62, OVCAR-3, OVCAR-5, U251, and SF-295, Jindal et al. [231] revealed that 17α-ethynylandrostene derivatives **248**–**250**, and estrane derivatives **251** and **252** displayed noticeable antiproliferative activities.

(6*E*)-Hydroximinocholest-4-en-3-one (**253**), a steroidal oxime previously reported from *Cinachyrella alloclada* and *C. apion* [232], was also shown to exhibit selective cytotoxic activity against several types of cancer cells, such as P-388, A-549, HT-29 (IC_50_: 1.25 g/mL), and MEL-28 tumor cells (IC_50_: 2.5 g/mL) [233]. Unfortunately, this compound and its analog (**254**) exhibited only a marginal activity against sk-Hep-1, H-292, PC-3, and Hey-1B [234].

Estrone-16-oxime ethers have also shown cytotoxicity toward human cancer cell lines HeLa, MCF7, A2780, and A431. Among them, **255** (IC_50_ 3.52, 4.13, and 4.61 µM) was primarily active against HeLa, MCF7, and A2780, respectively, while **256** (IC_50_ 5.63 and 13.25 µM) exerted more effects on HeLa and A431, respectively [235]. These estrone-16-oxime analogs were able to activate caspase-3 and change the mRNA level expression of endogenous factors regulating the G1-S phase transition (retinoblastoma protein, CDK4, and p16) [235]. Furthermore, several 6*E*-hydroximino-4-ene steroid oxime derivatives and their oxalate salts were evaluated for their in vitro antineoplastic activity against a panel of cancer cell lines. Many of these compounds showed moderate to good antiproliferative activity against leukemia, colon, melanoma, and renal cancer cell lines, with compounds **257** and **258** demonstrating the highest activity across all cell lines [236].

The anticancer activity of benzylidene pregnenolone oxime derivatives was also screened against HT-29, HTC-15, SF-295, HOP-62, A-545, and MCF-7 cancer cell lines. Compounds **259** (IC_50_ 2.44, 3.53, 3.79, 1.50, 1.42, and 1.90 μM), **260** (IC_50_ 2.43, 1.46, 8.86, 4.18, 1.97, and 1.34 μM), **261** (IC_50_ 2.35, 0.31, 1.67, 3.57, 3.56, and 0.60 μM), and **262** (IC_50_ 2.37, 0.65, 5.44, 0.81, 7.17, and 1.91 μM) exhibited significant cytotoxic activities against these cell lines.

Acharya and Bansal [236] also evaluated a group of structurally diverse steroidal oxime ethers bearing either a 3-, 6-, or 22-*O*-methyloxime or *O*-benzyloxime for their antiproliferative activities. Some of these compounds exhibited a distinct cytotoxicity against SGC-7901 (**263**: IC_50_ 17.3 μmol/L; **264**: IC_50_ 15.2 μmol/L), Bel-7404 (**265**: IC_50_ 14 μmol/L; **264**: IC_50_ 13.3 μmol/L), and CNE-2 (**265**: IC_50_ 14.1 μmol/L) cancer cell lines, with the study indicating that derivatives bearing 22-*O*-methyloxime and 22-*O*-benzyloxime exhibited better antiproliferative activities [236].

*D*-Secooxime derivatives of 13β- and 13α-estrone were shown to possess potent antiproliferative activity against a panel of human adherent cell lines (HeLa, MCF-7, A2780, and A431), with compounds **266** (IC_50_: 1.2 μM, 0.9 μM, 0.8 μM, and 2.6 μM), **267** (IC_50_: 1.7 μM, 0.7 μM, 0.9 μM, and 2.1 μM), and **268** (IC_50_: 0.8 μM, 0.4 μM, 0.9 μM, and 0.6 μM) exhibiting the best activity [237]. The *E* isomer of **266** (the *Z* isomer), which was also the most selective of the series, induced a cell cycle disturbance in the S phase [237]. Some 4-azasteroidal-20-oxime derivatives of progesterone also displayed anticancer activity in vitro against T24 cells, with compounds **269** (IC_50_: 1.90 μM), **270** (IC_50_: 1.94 μM), and **271** (IC_50_: 1.99 μM) being the most potent of the series [238]. This study indicated that methyl and ethyl oxime-ethers exhibited higher activity compared to the aryl-substituted oxime-esters [238].

Canário et al. [230] also studied several steroidal oximes of the estrane family, many of which exhibited interesting effects against the proliferation of various tumor cell lines when compared to the parent ketone compounds. Among these oximes, compound **272** displayed the highest activity against LNCaP cancer cells, along with a significant selectivity index. Furthermore, this compound induced cell cycle arrest in G2/M prophase and caused condensed and fragmented nuclei, characteristic of apoptosis [230]. In an e-screening assay, compound **272** also promoted the proliferation of T47-D cells, and docking studies indicated that this compound has relevant affinities for ERα and β-tubulin, providing a basis for its mechanism of action and its estrogenic effect. Interestingly, oximes bearing halogens in the A-ring, such as 2,4-diiodoestrone oxime **273** and 2,4-dibromoestrone oxime **274**, exhibited selectivity for HepaRG cancer cells [230]. Another A-ring functionalized derivative, 2-nitroestrone oxime (**275**) (but not its 2,4-dinitro analog), showed higher cytotoxicity against HepaRG and MCF-7 cancer cells [230].

While evaluating *O*-alkylated oxyimino androst-4-ene derivatives synthesized from the corresponding isomerically pure 3*E*-oximes in vitro for their cytotoxic activity against eight human cancer cell lines, as well as normal fetal lung (MRC-5) and human foreskin (BJ) fibroblasts for selectivity, most compounds demonstrated strong activity against the malignant melanoma (G-361), lung adenocarcinoma (A549), and colon adenocarcinoma (HT-29) cell lines [239]. Compound **276** emerged as the most promising candidate. These compounds also induce significant apoptosis in these cells, and further in vitro investigations against cytochrome P450 enzymes revealed their selective binding to the active sites of human steroid hydroxylases, including CYP7, CYP17A1, CYP19A1, and CYP21A2 [239].

Kolsi et al. [240] reported a series of dehydroabietic oxime derivatives with potential as antitumor agents, targeting pancreatic cancer and cancer-related inflammation. Compounds **277** and **278** were the most potent against human pancreatic cancer Aspc-1 cells, both exhibiting an IC_50_ value of 8.6 μM. Additionally, they demonstrated anti-inflammatory activity by inhibiting nitric oxide production, a crucial inflammatory mediator in the tumor microenvironment [240]. Further studies revealed that these compounds induced cancer cell differentiation and simultaneously downregulated cyclin D1 expression while upregulating p27 levels, consistent with cell cycle arrest in the G1 phase. Moreover, a kinase profiling study showed that some of these compounds exhibited isoform-selective inhibitory activity on RSK2 and AGC kinase, which are known to be implicated in cellular invasion and metastasis [240].

Both the androgen receptor (AR) and Cytochrome P450 17A1 (CYP17) are important targets for blocking androgen signaling. A steroidal imidazole oxime derivative (**279**) has demonstrated favorable biological profiles in both enzymatic and cellular assays [241]. This compound exhibited AR inhibition with an IC_50_ of 0.5 μM, which was 27-fold higher than that of the non-oxime parent compound, and a comparable CYP17 inhibition, with an IC_50_ of 11 μM [241]. Structure–activity relationship (SAR) studies revealed that introducing an oximino group at the C-3 position of the steroidal scaffold enhances AR antagonistic activity [241]. Additionally, **279** demonstrated promising antiproliferative effects on LNCap cell lines, with an IC_50_ value of 23 μM, superior to the positive control Flutamide (IC_50_ = 28 μM) [241].

Furthermore, molecular docking was employed to investigate the binding mode of compound **279** with both AR (PDB: 4oea) and CYP17 proteins (PDB: 3ruk). This compound precisely fits into the ligand binding domain (LBD) of AR [241]. Further analysis revealed that the hydroxyimino group at the C-3 position established hydrogen bonds with Gln-711, Met745, and Arg752, while the hydroxy group formed hydrogen bonds with Asn705 and Thr877 [241]. Additionally, the docking study with CYP17 provided evidence of its promising CYP17 inhibitory activity, as it positioned well within the active site of CYP17 as illustrated in Figure 16. The nitrogen atom of *N*-methylimidazole formed a hydrogen bond with the Arg239 amino acid residue, and the oxygen atom at the C-3 position interacted with the heme of CYP17, which is crucial for inhibiting CYP17 activity [241].

A series of oleanolic acid oxime derivatives and their conjugates with aspirin have also been investigated for their effects on the expression and activation of NF-κB in human hepatoma HepG2 cells [242]. The oleanolic acid oxime derivatives exhibited a stronger cytotoxic effect against HepG2 cells compared to their conjugates with aspirin. However, conjugation of oleanolic acid oxime with aspirin resulted in improved downregulation of NF-κB expression and activation [242]. Among the aspirin hybrids, compounds **280** (3-(2-acetoxy)benzoyloxyiminoolean-12-en-28-oic acid morpholide) and **281** (3-(2-acetoxy)benzoyloxyiminoolean-12-en-28-oic acid methyl ester), differing by the presence of morpholide and methyl ester groups at the C-17 position of the oleanolic acid molecule (Figure 17), were the most potent [242]. These oleanolic acid oxime derivatives, notably their conjugates with aspirin, downregulated the expression of COX-2 in HepG2 cells by modulating the NF-κB signaling pathway, suggesting their potential application in preventing liver inflammation and cancer [242].

Analogs of natural 5*Z*,9*Z*-dienoic acids, hybrid molecules based on the oximes of cholesterol, pregnenolone, and androsterone with 1,14-tetradeca-5*Z*,9*Z*-dienedicarboxylic acid were synthesized and evaluated, along with their parent oximes, for in vitro antitumor activity against HeLa, Hek293, U937, Jurkat, and K562 cell lines [243]. Compounds **282** (IC_50_: 0.82, 1.01, 0.74, 0.62, and 0.64 μM), **283** (IC_50_: 0.67, 0.53, 0.45, 0.74, and 0.71 μM), **284** (IC_50_: 0.68, 0.27, 0.38, 0.49, and 0.30 μM), **285** (IC_50_: 0.51, 0.33, 0.49, 0.76, and 0.58 μM), **286** (IC_50_: 0.32, 0.34, 0.41, 0.13, and 0.29 μM), and **287** (IC_50_: 0.02, 0.02, 0.04, 0.03, and 0.17 μM) demonstrated strong inhibition of tumor viability in all the tested cell lines, respectively [243]. Flow cytometry assays indicated that these molecules efficiently induced apoptosis in HeLa, Hek293, U937, Jurkat, and K562 cells [243].

While evaluating a series of androstene oxime-nitrogen mustard bioconjugates for their cytotoxicity against a panel of 60 cancer cell lines, 17*E*-steroidal oxime-benzoic acid mustard ester 3β-acetoxy-17*E*-[*p*-(*N*,*N*-*bis*(2-chloroethyl)amino)]benzoyloxy-imino-androst-5-ene (**288**) emerged as the most potent conjugate [244]. It exhibited significant growth inhibition on the IGROV1 ovarian cancer cell line, with a GI_50_ of 0.937 µM. Generally, the androstene oxime-nitrogen mustard conjugates derived from the D-ring of the steroid skeleton displayed superior antineoplastic activity compared to those derived from other rings [244].

Bu et al. [245] evaluated a series of designed and synthesized 5α,8α-epidioxyandrost-3β-ol-17-(*O*-phenylacetamide)oxime derivatives for their antiproliferative activity against human hepatocellular carcinoma cells (HepG2, Sk-Hep1) and human breast cancer cells (MCF-7, MDA-MB231). Compounds **289** (IC_50_: 12.14, 14.63, 20.36, and 15.82 µM), **290** (IC_50_: 9.30, 12.12, 22.74, and 17.53 µM), and **291** (IC_50_: 14.84, 11.08, 19.50, and 14.25 µM) exhibited moderate in vitro antiproliferative activities. Additionally, fluorescence imaging revealed that the coumarin-**289** conjugate (**292**) primarily accumulates in the mitochondria, resulting in a stronger anticancer activity compared to the parent structure [245].

Photodynamic therapy is progressively emerging as an alternative treatment for both malignant and non-malignant cancers, despite its limitations in targeting specificity, which is critical for preventing systemic toxicity. Pavlíčková et al. [246] reported the synthesis and biological evaluation of a bimodal oxime conjugate (**293**) encompassing a photosensitizing drug (red-emitting pheophorbide a) and nandrolone, a steroid that specifically binds to the androgen receptor, often overexpressed in various tumors. Since light-triggered therapies could potentially have a significantly positive impact on the treatment of hormone-sensitive cancers, this targeted bimodal oxime conjugate was evaluated on prostatic cancer cell lines in vitro, using an AR-positive cell line (LNCaP) and an AR-negative/positive cell line (PC-3) [246]. U-2 OS cells, both with and without stable AR expression, were used as a second cell line model. This conjugate not only exhibited photodynamic activity and AR specificity, but also demonstrated a more pronounced phototoxic effect compared to pristine pheophorbide a. Live-cell fluorescence microscopy provided clear evidence that the **293** localizes in the endoplasmic reticulum and mitochondria [246]. A competitive localization study with an excess of nonfluorescent nandrolone allowed for the displacement of this fluorescent conjugate (**293**) from the cell interior, providing additional evidence for its binding specificity. Figure 17 summarizes some examples of anticancer oleanolic acid oxime derivatives and steroidal oxime conjugates.

### 2.8. Anticancer Activity of Oxime Liganded Metals and Cyanoximates Metallic Complexes

The clinical success of the anticancer cisplatin, discovered in the 1960s, stimulated a significant interest in the use of metal complexes in cancer treatment [247,248]. As a matter of fact, cisplatin is believed to be instrumental in the cure of over 90% of testicular cancer cases, in addition to playing an essential role in the treatment of other cancers, including melanomas, lymphomas, as well as ovarian, bladder, and cervical cancers [247]. To overcome the two main problems associated with cisplatin treatment, namely severe side effects and resistance, a number of approaches for the design and implementation of platinum-based drugs capable of being activated on-site in the tumor tissue have been developed. For example, classes of kinetically inert platinum-based anticancer prodrugs, suitable for on-site transformation to more reactive platinum(II) species, either leveraging the reducing environment in solid tumors, or activated on site through phototherapy, have been developed [249,250].

Furthermore, *trans*-configured oxime-liganded platinum complexes, which displayed many attributes different from those of established anticancer agents, including a distinct DNA binding behavior, increased cellular accumulation, and an unusual pattern of protein interaction, have gained traction as a new class of anticancer therapies [248,249,251]. As a result, a number of platinum(II) oxime complexes, inert at physiological pH, but capable of being activated into their cytotoxic platinum forms under a slightly acidic pH environment, have been developed [252,253,254,255,256], and some illustrative examples are provides in Figure 18.

Some of these kinetically inert platinum(II) complexes bearing a chelating oxime ligand were evaluated for their cytotoxic properties on three human tumor cell lines originating from ovarian carcinoma (CH1), colon carcinoma (SW480), and non-small cell lung cancer (A549) [255]. This evaluation included a comparative investigation under standard pH screening conditions (pH 7.4) as well as in an acidified medium (pH 6.0), with the goal of determining their pH-dependent activation properties [255]. Among these compounds, **294** (IC_50_ 3.9 μM), **295** (IC_50_ 16 μM), **296** (IC_50_ 1.0 μM) were primarily active against CH1 cell line, while **297** (IC_50_ 8.3 and 12 μM) showed potency against both CH1 and SW480, respectively [255]. Compound **298** was the most potent of the series across the board, exhibiting IC_50_ values of 1.1, 0.97, and 5.6 μM against CH1, SW480 and A549, respectively [255].

Bartel et al. [257] reported cellular accumulation experiments and in vitro DNA platination studies involving three pairs of *cis*- and *trans*-configured acetone oxime platinum(II) complexes, as well as one pair of pentan-3-one oxime platinum(II) complexes. They found that, overall, *trans*-platinum(II) oxime complexes, **299**(*t*) (IC_50_ 0.22 µM), **300**(*t*) (IC_50_ 2.5 µM), **301**(*t*) (IC_50_ 2.8 µM), and **302**(*t*) (IC_50_ 1.9 µM), exhibited higher cytotoxicity compared to their corresponding *cis* congeners: **299**(*c*) (IC_50_ 3.4 µM), **300**(*c*) (IC_50_ 45 µM), **301**(*c*) (IC_50_ 1.8 µM), and **302**(*c*) (IC_50_ 15 µM), on the cisplatin-resistant SW480 cell line [257]. This study also revealed that the concentration of *trans*-configured complexes within SW480 cells was up to 100 times higher than that of cisplatin and up to 50 times higher than their *cis*-configured counterparts. Furthermore, the ‘r’ values (number of platinum atoms per nucleotide) were more than ten-fold higher in cells treated with *trans* complexes compared to those treated with cisplatin [257].

The interaction of these complexes with DNA was further investigated in cell-free experiments using plasmid DNA (pUC19), in capillary zone electrophoresis with the DNA model 2-deoxyguanosine 5′-monophosphate, and through in vitro experiments measuring the extent of DNA damage in the comet assay. While incubation with *cis* complexes did not induce any DNA degradation, the *trans* complexes resulted in pronounced strand cleavage [257]. The most likely explanation for the difference in behavior between these complexes and the influence of their configuration is probably the formation of platinum–DNA mono-adducts in the case of the *trans* complexes, which potentially contributed to their cytotoxic effect [257].

The cytotoxicity of some *trans* platinum(II) oxime complexes and their head-to-tail-oriented oximato-bridged dimers featuring a six-membered diplatinacycle and representing a unique class of oxime ligands coordinated to platinum via an oxygen atom, was also evaluated on three human cancer cell lines, including the cisplatin-sensitive CH1 cells as well as cisplatin-resistant SW480 and A549 cells [256]. All these compounds, including **303** (IC_50_ 6.6, 3.5, and 14 μM), **304** (IC_50_ 3.3, 3.3, and 13 μM), **305** (IC_50_ 6.2, 3.6, and 21 μM), **306** (IC_50_ 3.4, 3.3, and 13 μM), **307** (IC_50_ 0.95, 1.2, and 12 μM), **308** (IC_50_ 1.8, 1.9, and 12 μM), **309** (IC_50_ 0.94, 1.0, and 5.1 μM), **310** (IC_50_ 0.96, 1.3, and 5.7 μM), and **311** (IC_50_ 2.2, 2.5, and 9.5 μM), exhibited strong potency against all the three cell lines [256]. The dimerization resulted in a significant (up to seven-fold) improvement of IC_50_ values of (aldoxime)Pt(II) compounds, with the reverse trend observed for analogous complexes featuring ketoxime ligands. Notably, the dimers did not produce any cross-resistance with cisplatin in SW480 cells, as it demonstrated an up to two-fold increase in cytotoxicity in comparison with its effect on the CH1 cell line [256]. Furthermore, one monomer **305** as well as one dimer **309** proved to be strong apoptotic agents capable of inducing cell death even in the cisplatin-resistant SW480 cell line [256]. Increasing the alkyl chain length of the oxime ligand using propionaldehyde oxime (**305**, **306**, **309**, and **310**) instead of acetaldehyde oxime (**303**, **304**, **307**, and **308**) did not result in a significant increase in cytotoxicity. However, the *E*- and *Z*- configuration of the coordinated aldoximes appeared to have some impact on the obtained IC_50_ values, as monomeric complexes **304** and **306** featuring both *E*-and *Z*-aldoximes exhibited an up to two-fold increase in cytotoxicity against CH1 cells compared to their isomers **303** and **305**, containing two *Z* ligands [256].

Pt(II) complexes bearing triphenylphosphine and chelating oximes were found to efficiently penetrate resistant cells and induce significant antiproliferative effects, despite minimal interaction with DNA [258]. Among them, compounds **312** with IC_50_ values of 3.87, 5.18, 3.83, 14.1, 0.62, 1.43, and 3.15 μM, and **313** with IC_50_ values of 2.47, 2.50, 2.63, 3.61, 0.31, 1.08, and 2.60 μM, exhibited potency against a panel of human tumor and mesothelial cells, including HeLa, A549, HT-29, MSTO-211H, A2780, A2780cis, and Met-5A, respectively [258]. Notably, compound **313** exhibited an outstanding cytotoxicity in ovarian carcinoma cells (A2780), with a significant and unexpected antiproliferative effect on cisplatin-resistant cells (A2780cis). Further studies into the intracellular mechanism of action indicated that **313** had a lower ability to bind to DNA compared to cisplatin (used as a control), but demonstrated a noticeably higher uptake in resistant cells [258]. A higher accumulation in the mitochondria, added to its ability to induce a concentration-dependent mitochondrial membrane depolarization and a strong stimulation of the production of intracellular reactive oxygen species, suggested a mitochondrion-mediated pathway as responsible for the intriguing cytotoxic profile of complex **313** [258].

Eddings et al. [259] reported the synthesis of bivalent palladium and platinum cyanoximates and evaluated their in vitro antiproliferative activity against human cervical cancer HeLa cell lines, using cisplatin as a positive control. Among these compounds, **314** (Figure 18) and **315** (Figure 19) were found to be active, causing the death of 16% and 28% of the cells, respectively, compared to a 55% reduction observed for cisplatin under the same conditions [259].

As the global annual sales of platinum-based anticancer drugs exceeds two billion U.S. dollars [251], a considerable number of other metal complexes has been prepared and evaluated as potential chemotherapeutic agents as illustrated in Figure 19, although only a few have entered clinical trials [260,261].

Oxime-based palladacycles were also found to exhibit significant cytotoxic activity against the HT29, A549, and HeLa cancer cell lines [262,263]. Among them, the dinuclear complex **316** displayed greater anticancer activity in comparison to the mononuclear complexes **317** and **318**. All these compounds acted in a time- and dose-dependent manner, resulting in a significant decrease in cell viability when compared to the control group in all three cell lines for each of the complexes [263].

Half-sandwich ruthenium, rhodium, and iridium complexes synthesized from aldoxime, ketoxime, and amidoxime ligands also exhibited in vitro antitumor activity against HT-29, BE, and MIA PaCa-2 cell lines, especially for complexes **319** and **320 [264]**. Complex **319** demonstrated comparable activity but greater selectivity for MIA PaCa-2 cells compared to cisplatin. Further studies have revealed that complexes **321**, **322**, **323**, and **324** induced significant apoptosis in Dalton’s ascites lymphoma (DL) cells, with **319** showing statistically significant in vivo activity in DL-bearing mice.

While investigating the ability of water-soluble chiral arene Ru(II) complexes bearing amino-oxime ligands to interact with plasmid (pBR322) DNA and to exhibit cytotoxic and antimetastatic effects in vitro against androgen-unresponsive human prostate cancer PC3 cell lines, Benabdelouahab et al. [265] discovered that these compounds significantly reduced cell viability in PC3 cells compared to control, with an inhibitory activity of 75–82% at concentrations of 50 μM. Compounds **325** (IC_50_ 24.4 μM), **326** (IC_50_ 14.8 μM), and **327** (IC_50_ 21.5 μM) showed considerably higher cytotoxicity than cisplatin (IC_50_ 510 μM) under the same experimental conditions [265].

Furthermore, Saad El-Tabl et al. [266,267] and Samy and Shebl [268] reported a panel of metal complexes derived from 2-hydroxy-3-(hydroxyimino)-4-oxopentan-2-ylidene)benzohydrazide and (*E*)-3-(2-(5, 6- diphenyl-1,2,4- triazin-3- yl)hydrazono)butan-2- one oxime, respectively. These compounds included iron (**328**), silver (**329**), zirconium (**330**), uranium (**331**), cobalt (**332**), and copper (**333**, **334**, **335**, and **336**) complexes, all of which were evaluated for their anticancer effects. Among them, copper complexes, including **334** (IC_50_ 6.49 μM), **335** (IC_50_ 2.67 μM), and **336** (IC_50_ 2.24 μM), as well as the cobalt complex **332** (IC_50_ 36.8 μM), exhibited noticeable antiproliferative potency against HepG2 human liver cancer cell lines [266,267].

## 3. Conclusions

This report represents one of the most comprehensive assessments of the anticancer potential of structurally diverse molecular frameworks bearing oxime and oxime ether moieties. Specific examples have been carefully selected to illustrate the distinct contributions of these functional groups to general strategies for molecular design, modulation of biological activities, computational modeling, and structure–activity relationship studies. The primary focus is on the role of this functional group in enhancing the physicochemical and anticancer properties of the resulting molecules. Additional mechanistic investigations, molecular dynamics, and docking studies indicate that the mechanism and the potential drug targets of the resulting compounds also depend on the chemical framework to which the functional group is attached. In some cases, the oxime or oxime ether groups enhanced the already available anticancer properties, while in others, these groups were fully responsible for the observed properties, as without this functional group this scaffold was completely inactive. The pharmacokinetic and pharmacodynamic profiles, as well as drug-likeness, also appeared to be impacted to a greater extent by the addition of the oxime or oxime ether functional group. Regardless of the situation, these compounds appear to inhibit, interact with, or interfere with a number of key markers in cancer pathogenesis and malignancy. As a result, it is not unlikely that some of these oxime and/or oxime ether-bearing molecular frameworks could soon advance to clinical trials or even make it to the market.

## Figures and Tables

**Figure 1 ijms-24-16854-f001:**
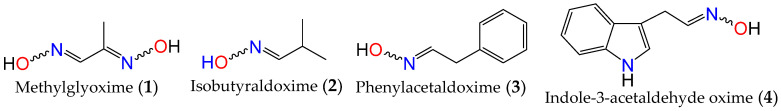
First synthetic and earliest amino acid-derived oximes from plants [33,34,35,36].

**Figure 2 ijms-24-16854-f002:**
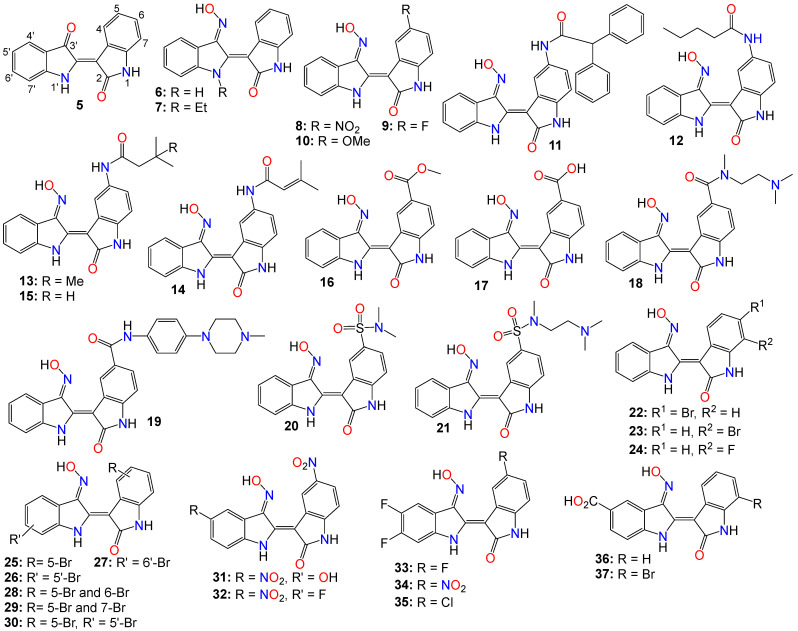
Illustrative examples of anticancer indirubin-3′-oxime derivatives, bearing a diversity of substitution patterns on the core bis-indole scaffold.

**Figure 3 ijms-24-16854-f003:**
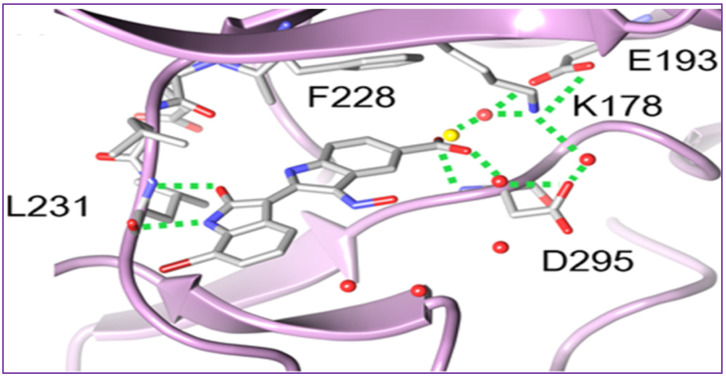
Crystal structure of the **37**-DYRK2 complex revealing a nontypical binding mode, in which the orientation of the indirubin core is in good accordance with the predicted binding mode [94] (https://pubs.acs.org/doi/10.1021/ml300207a), accessed on 27 August 2023. This image is reused with the express permission of ACS, and any further permissions related to this material should be directed to ACS.

**Figure 4 ijms-24-16854-f004:**
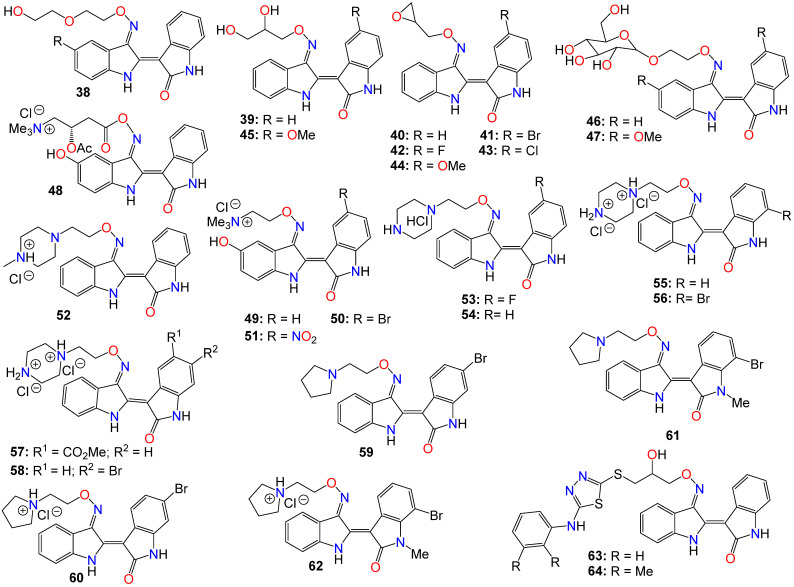
Illustrative examples of anticancer indirubin-3′-oxime ether derivatives.

**Figure 5 ijms-24-16854-f005:**
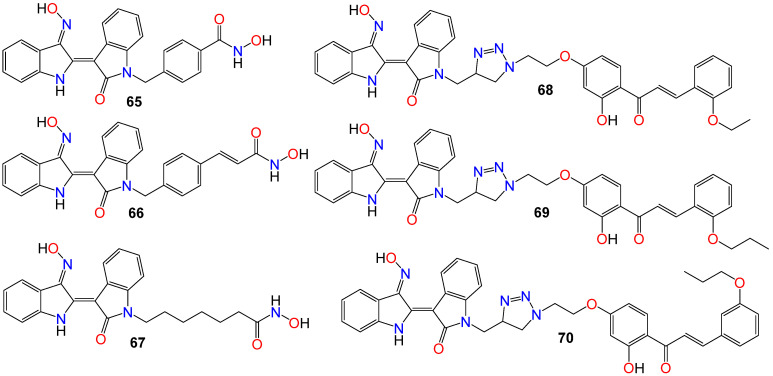
Illustrative examples of anticancer indirubin-3′-oxime derivative with substituents at *N*^1^ of the bisindole core.

**Figure 6 ijms-24-16854-f006:**
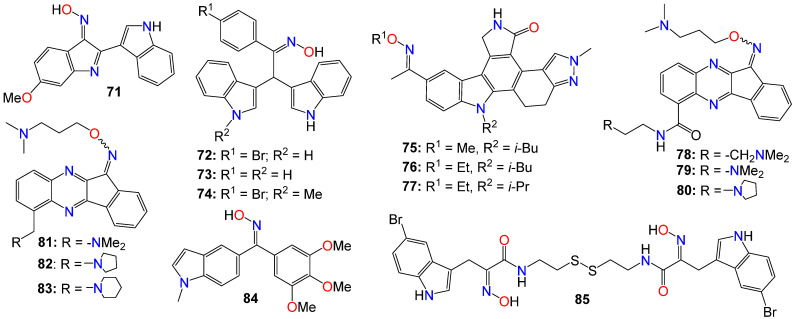
Illustrative examples of other antitumor indole derivatives bearing an oxime moiety.

**Figure 7 ijms-24-16854-f007:**
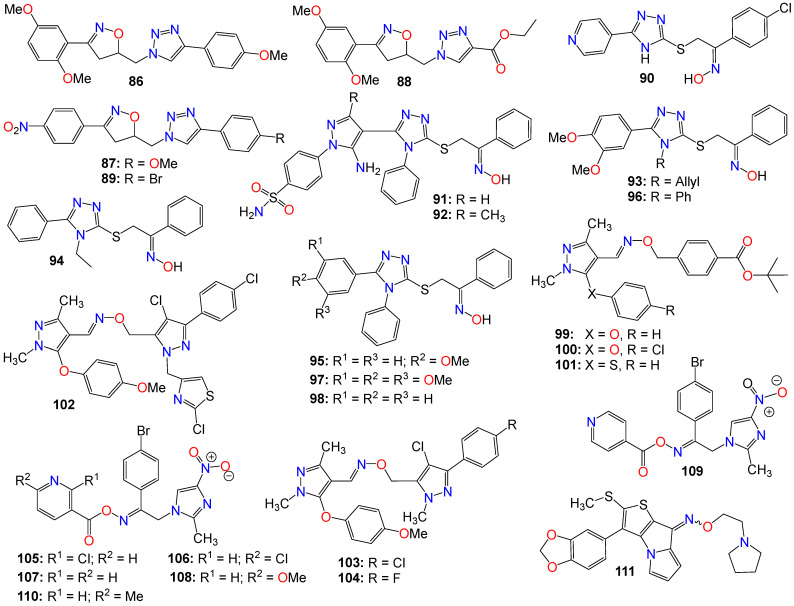
Illustrative examples of anticancer triazole- and pyrazole-based oximes and oxime ethers.

**Figure 8 ijms-24-16854-f008:**
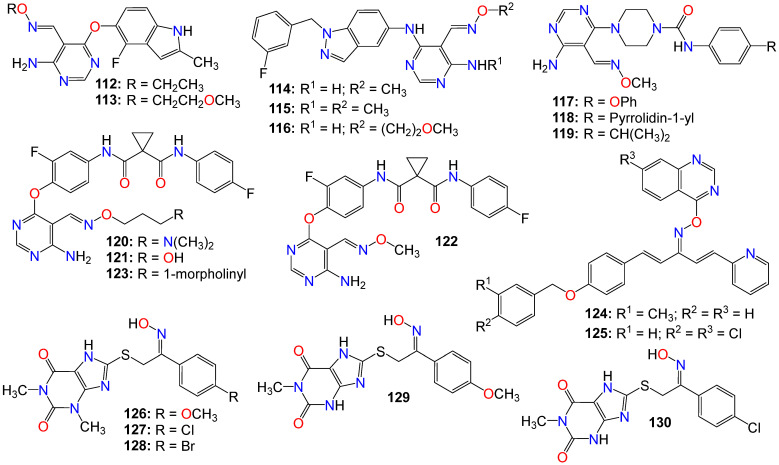
Illustrative examples of miscellaneous small nitrogen-containing hetorocyclic-based oximes with anticancer properties.

**Figure 9 ijms-24-16854-f009:**
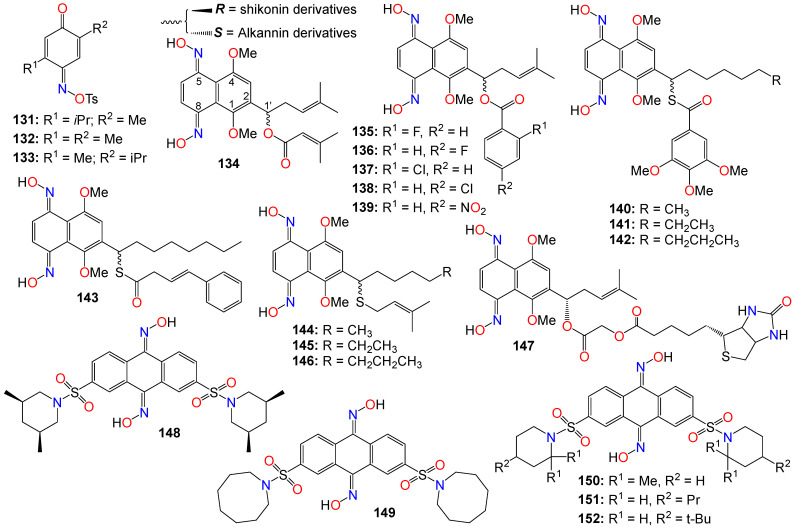
Illustrative examples of anticancer quinone-based oximes and oxime ethers.

**Figure 10 ijms-24-16854-f010:**
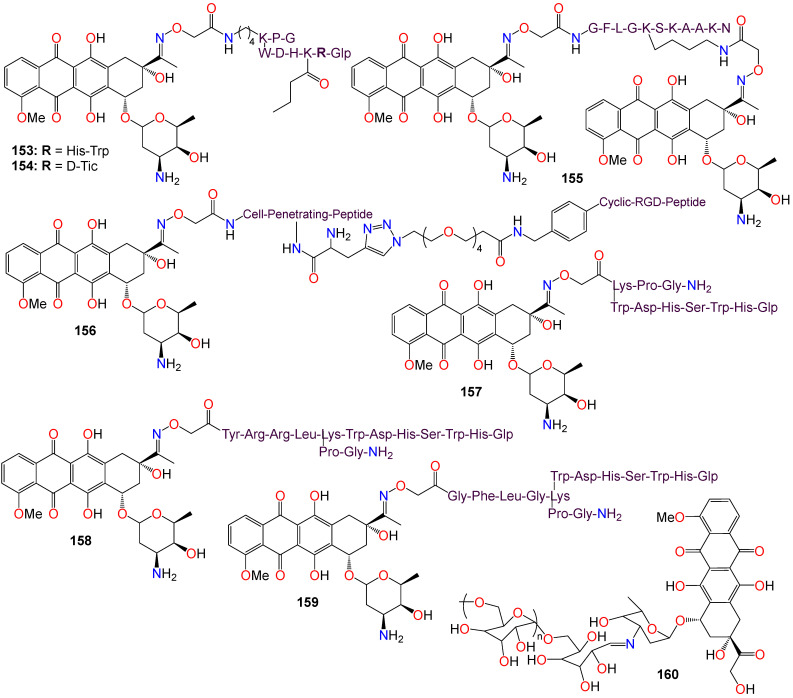
Illustrative examples of anticancer quinone oxime bond-linked bioconjugates.

**Figure 11 ijms-24-16854-f011:**
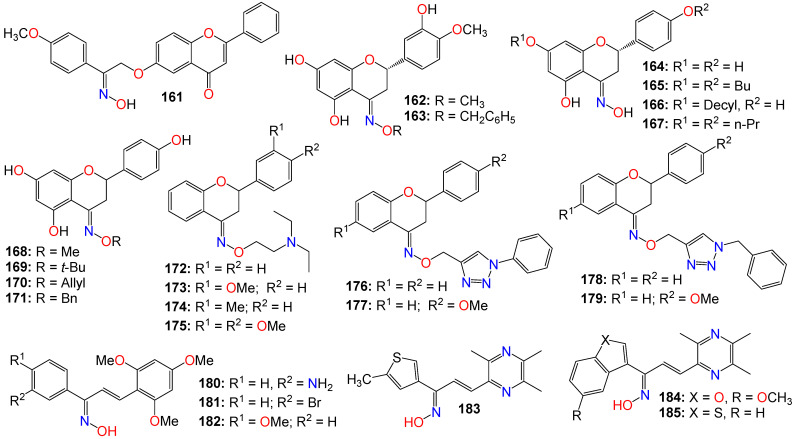
Illustrative examples of anticancer flavonoid-based oximes and oxime ethers.

**Figure 12 ijms-24-16854-f012:**
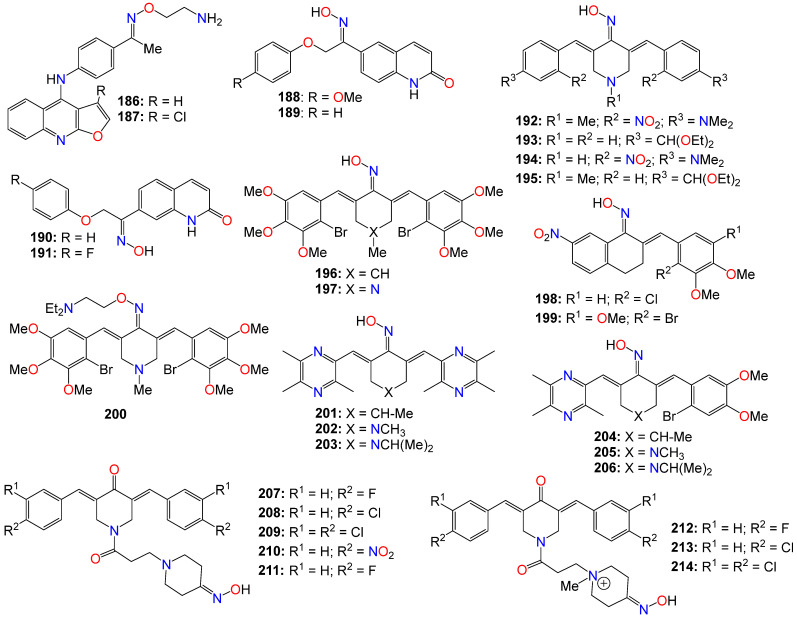
Illustrative examples of anticancer quinolinone-, cyclohexanone-, piperidin-4-one- and tetralone-based oximes.

**Figure 13 ijms-24-16854-f013:**
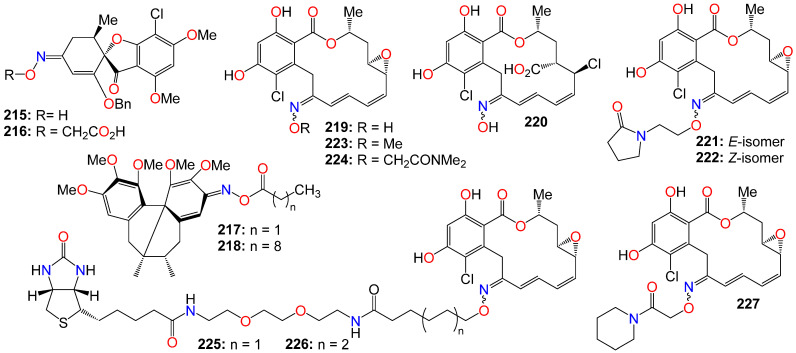
Illustrative examples of radicicol oxime and oxime ether derivatives with antitumor activities.

**Figure 14 ijms-24-16854-f014:**
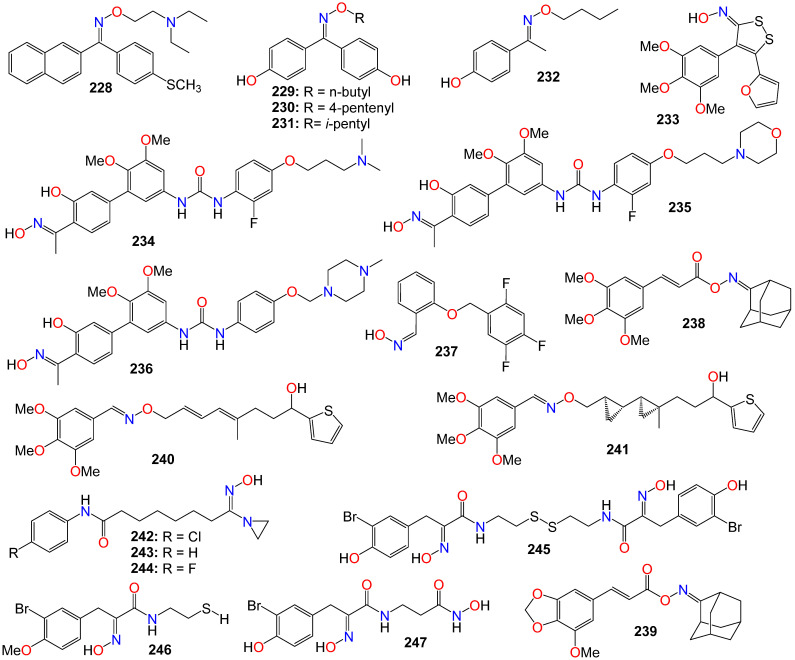
Illustrative examples of anticancer small aromatic and phenolic-based oximes and oxime ethers.

**Figure 15 ijms-24-16854-f015:**
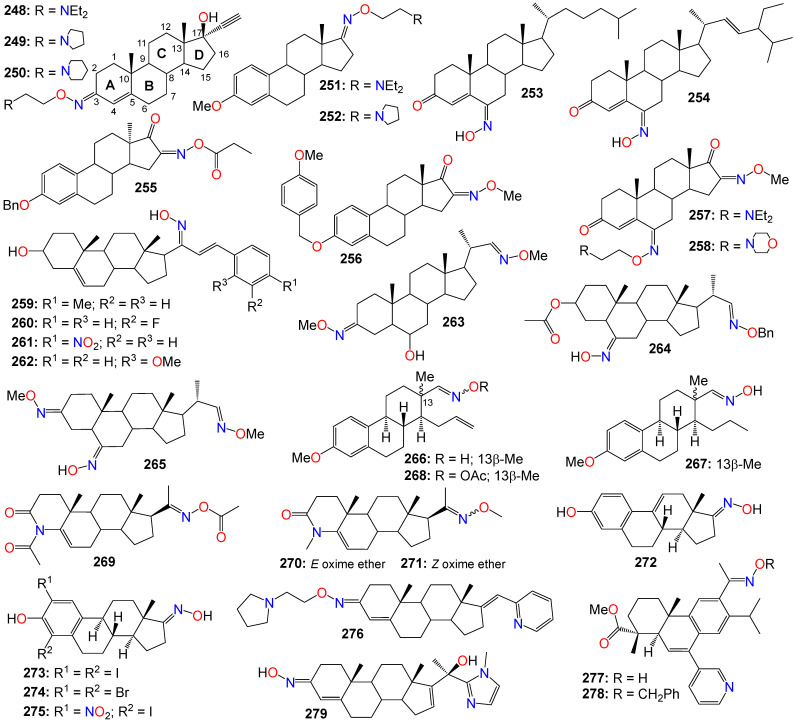
Illustrative examples of anticancer steroidal oximes and oxime ethers.

**Figure 16 ijms-24-16854-f016:**
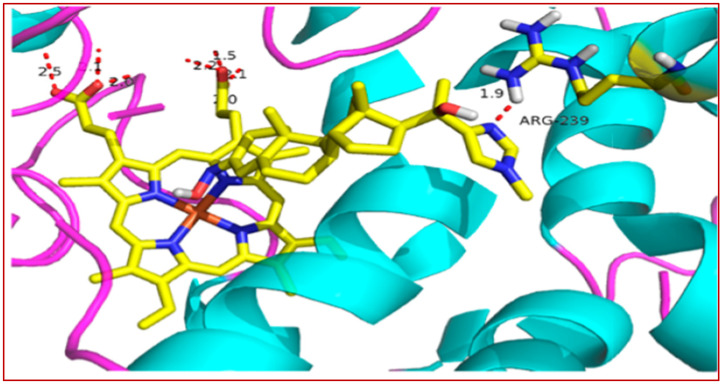
The binding mode of **279** docked into the active site of CYP17, with the red dotted line representing a hydrogen bond [241] (https://doi.org/10.1016/j.steroids.2019.03.003 accessed on 27 August 2023). This image was reused with express permission from Elsevier, and any further permissions related to this material should be directed to Elsevier.

**Figure 17 ijms-24-16854-f017:**
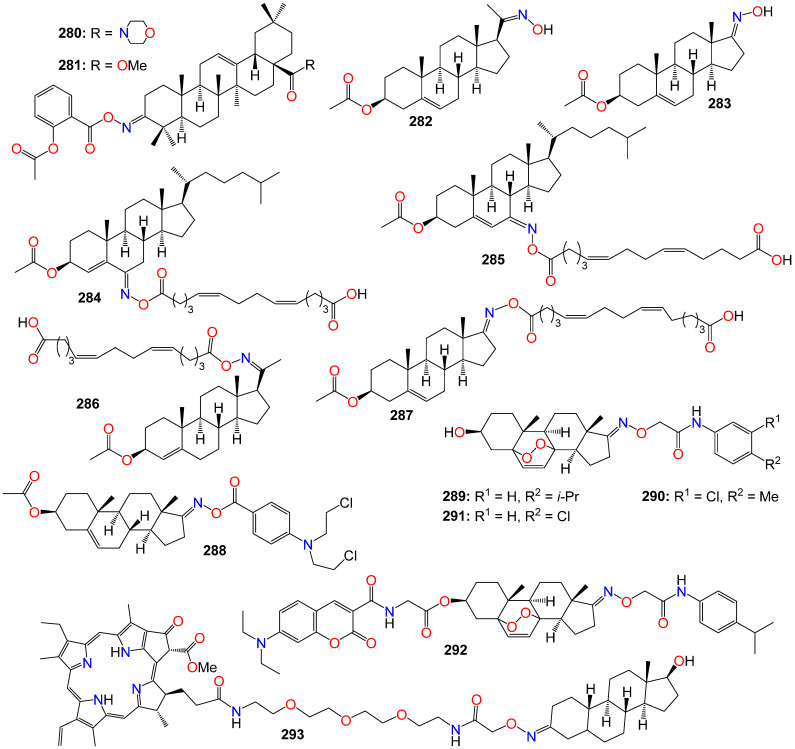
Illustrative examples of anticancer oleanolic acid oxime derivatives and steroidal oxime conjugates.

**Figure 18 ijms-24-16854-f018:**
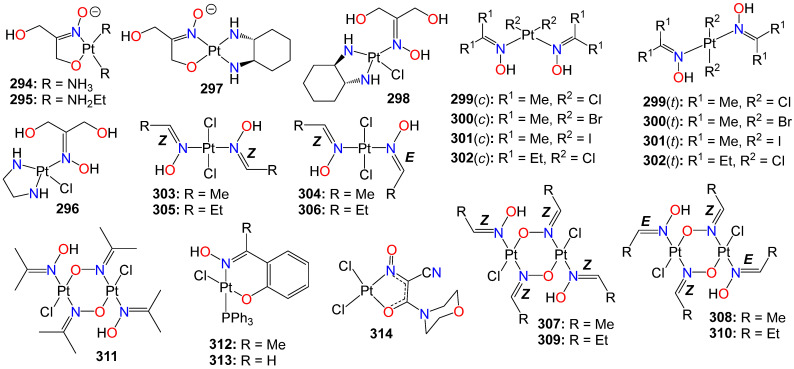
Illustrative examples of anticancer platinum complexes bearing chelating oxime ligands.

**Figure 19 ijms-24-16854-f019:**
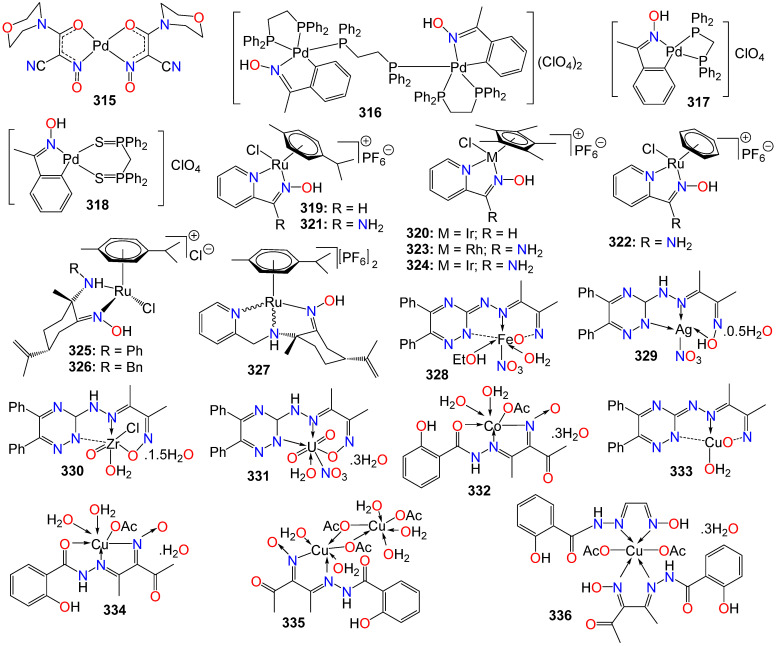
Illustrative examples of miscellaneous oxime-chelated metal complexes as potential chemotherapeutic agents for cancer treatment.

## Data Availability

All needed information is contained within the manuscript, no supplemental data available.

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
