# Peer review of "The Fundamental Role of Oxime and Oxime Ether Moieties in Improving the Physicochemical and Anticancer Properties of Structurally Diverse Scaffolds"

_ijms, 2023, doi:10.3390/ijms242316854_

Round 1

Reviewer 1 Report

Comments and Suggestions for Authors

This paper describes the activities of oxime containing molecules, in particular theirs anticancer activities. The amount of data is quite impressive (compounds and references) but classified depending on the scaffold. The text is well written and understandable, even though there are 60 pages and about 31 000 words!

The effect of the oxime group on activity is stated for some compounds (as line 502, 544, 1007, 1088, 1155, 1162, … for examples), but for many of the oxime compounds cited in this paper we do not know what the contribution of the oxime group is, in comparison to the other substitutions. With all these modifications, the contribution of the oxime group for many compounds become less clear for the SAR study and only few compounds have a comparison with vs without oxime group. Just an example: In the sentence line 489 “Docking studies indicated that compound 68 forms three hydrogen bonds towards GSK- 489 3β via two residues, namely Val135 and Thr138 [118].”. OK, but is the oxime group implicated or not in these bonds? If we remove the oxime group, do these bonds still exist? And the activity? This is this kind of questions that readers can wonder for many of the cited compounds when the comparison with or without oxime is not stated.

As a chemist, I am a little frustrated that no reaction can be found inside this paper (like a general reaction of formation of oxime and oxime ether). However, I think that this review can interest many readers more focused on SAR study than on synthesis itself. For this reason, I think that this paper is suitable for publication in IJMS without notable modification (except the few flaws below).

Flaws to fix:

172 "13 (IC50 14nM". Missing space.

269 "with this 5-cholro analog"

592 “hetorocyclic”

1135 " hydroxylamine moiety": oxime?

1396 " Yanmin et al. [236] ": There is not “Yanmin” in reference 236 (nor in the whole paper). Please check this reference and this line.

1624 “Among these compounds, 314 and 315 were found”: 314 is in Figure 18 and 315 in Figure 19. Please write the corresponding figure after each compound number. MDPI is quite strict about the figure being close to the text first describing the compound, but Figure 18 is for Pt and Figure 19 for other metals, so it should be fine like that.

1658 “Fatma and Magdy [268]”: It seems Fatma and Magdy are the first names, not the last names. The last names are correct in the reference section.

Pleased check that the reference are complete. For example, 263 “Appl Organomet Chem. 2019;33(9):n/a”: n/a should be e5098. Reference 264 “Appl Organomet Chem. 2017;31(7):n/a”: e3640.

Author Response

We would like to express our sincere appreciation for the invaluable comments and recommendations provided by the reviewers, which have greatly contributed to enhancing its overall quality of the manuscript.

It is important to clarify from the onset that oximes are not a family of compounds per se, but rather to a functional group that can be readily attached to a diverse range of chemical frameworks. The impact of this functional group on the anticancer properties of the resulting molecules varies significantly depending on the scaffold to which it is attached. Therefore, we have chosen to organize this manuscript based on chemical frameworks, believing this to be the most suitable approach given the diversity of structural frameworks involved.

We are particularly pleased that Reviewer 1 found the report insightful, well-written, and understandable, as addressing these aspects was a primary challenge throughout the entire process.

The main concern raised by this reviewer pertains to the lack of clarity regarding the contribution of the oxime group in many of the cited compounds. While this is a fair and understandable point, it is relevant primarily when considering their effects on each molecule or group of molecules in isolation. It is uncertain whether the overall effect of this functional group is fully understood in any of the studies summarized in this report. However, when examining the manuscript as a whole, it becomes apparent that oximes and oxime ethers are employed in various ways to enhance the anticancer properties of different scaffolds. In some cases, these functional groups are solely responsible for the observed activity or are clearly involved in the mechanism of action, while in other instances, their contribution is more subtle, such as improving the lipophilic balance or the pharmacodynamic and pharmacokinetic properties of the resulting molecules, serving as junction points in conjugates or act as ligands in metal complexes. Regardless of the scenario, it is crucial to consider the manuscript as a comprehensive entity to fully appreciate the diverse aspects and contributions of these functional groups to the antitumor activities of the resulting molecules.

A specific example use by the reviewer was related to the “docking studies indicating that compound 68 forms three hydrogen bonds towards GSK- 489 3β via two residues, namely Val135 and Thr138 [118].” The reviewer was wondering what we if the oxime group was implicated or not in these bonds? If the oxime group is removed, do these bonds still exist? And the activity?

To be candid, I had hoped that these studies would offer conclusive answers to these questions. However, it is important to acknowledge that this report does not necessarily have to encompass all the answers to be considered valuable. The mere fact that it raises these questions from the reviewer could spark a scientific debate necessary to delve into various aspects of oxime functionalities in anticancer molecules. As such, we remain optimistic that this report, at the very least, equips readers with the resources needed to cultivate a deeper understanding of the intricate relationship between the introduction of oxime and the resulting functionality, as we have been deliberate in highlighting any useful insights we were able to uncover throughout the course of this process.

 As a chemist, I am a little frustrated that no reaction can be found inside this paper (like a general reaction of formation of oxime and oxime ether).

As a medicinal chemist, I share the frustration expressed by this reviewer regarding the absence of chemical reactions in this report. However, I am confident that the reviewer would concur with me on the rather mundane nature of oxime chemistry, as they can be routinely obtained from any given amine and a hydroxylamine under trivial conditions. The inclusion of any reactions would have undoubtedly focus on the synthesis of the scaffold they are attached to, rather than on the preparation of the oximes themselves, and would have unnecessarily extended this already lengthy report.

As for the typos and grammatical errors

172 "13 (IC50 14nM". Missing space – corrected.

269 "with this 5-cholro analog" – corrected

592 “hetorocyclic” – corrected

1135 " hydroxylamine moiety": oxime? – it should have been oxime and it now corrected

1396 " Yanmin et al. [236] ": There is not “Yanmin” in reference 236 (nor in the whole paper). Please check this reference and this line. – It should Acharya and Bansal, now corrected

1624 “Among these compounds, 314 and 315 were found”: 314 is in Figure 18 and 315 in Figure 19. Please write the corresponding figure after each compound number. MDPI is quite strict about the figure being close to the text first describing the compound, but Figure 18 is for Pt and Figure 19 for other metals, so it should be fine like that. – corrected

1658 “Fatma and Magdy [268]”: It seems Fatma and Magdy are the first names, not the last names. The last names are correct in the reference section. – Should have been Samy and Shebl, and it is now corrected

Pleased check that the reference are complete. For example, 263 “Appl Organomet Chem. 2019;33(9):n/a”: n/a should be e5098. Reference 264 “Appl Organomet Chem. 2017;31(7):n/a”: e3640. – corrected 

Reviewer 2 Report

Comments and Suggestions for Authors

The review by Fotie et al. is interesting, very well written and very informative. As the authors claim themselves, the review is quite extensive and extremely long, which is not bad, but the way it is structured makes it quite difficult to read and follow. What I miss in this review are proper figures that illustrate the effects of these compounds on different proteins and signaling pathways related to cancer and figures (or tables) that summarize all the information in the text in a more concise way. The only figures in this review are of the chemical structures of the compounds and, in my view, some of those can be moved to Supplementary information. Another possible improvement would be the division of the text into more (sub-) chapters because right now a reader gets really confused by the immense amount of information provided. Finally, I am also missing proper conclusions. For such a long review the conclusions should be stated throughout the text (maybe at the end of each chapter). The current conclusions are quite general and since the review is so long, it is not very clear where they come from. I am sure it is quite to for the authors, but it is certainly not the case for the readers. Thus, I would suggest the authors to revise their review and present it in a more readable way before I can recommend it for publication in IJMS.

Author Response

The review by Fotie et al. is interesting, very well written and very informative. As the authors claim themselves, the review is quite extensive and extremely long, which is not bad, but the way it is structured makes it quite difficult to read and follow. What I miss in this review are proper figures that illustrate the effects of these compounds on different proteins and signaling pathways related to cancer and figures (or tables) that summarize all the information in the text in a more concise way.

This feedback is entirely reasonable, and I believe the reviewer would concur that no table could adequately encapsulate the intricacies of the interactions alluded to in this context. To address this, we made a deliberate effort throughout the text to delve deeper into these interactions. Given the multitude of docking studies condensed in this report, providing a figure for each would have turned this review into a collection of images. Therefore, we chose two figures that intrigued us the most and incorporated them into the manuscript.

It's worth noting that not many original papers include such images in their reports either. Since this paper is conceived to serve as a condensed overview of these studies, references are included at each step, to enable readers interested in more details from a specific study to access the original reports.

The only figures in this review are of the chemical structures of the compounds and, in my view, some of those can be moved to Supplementary information.

The incorporation of chemical structures within the text, beyond merely providing additional details about the nature of the scaffold under discussion, also facilitates a more straightforward understanding of the described structure-activity relationship. These structures are also intended to serve as additional resources for those not familiar with these chemical frameworks. As such, transferring these structures to supplemental information could be counterproductive.

Another possible improvement would be the division of the text into more (sub-) chapters because right now a reader gets really confused by the immense amount of information provided. Finally, I am also missing proper conclusions. For such a long review the conclusions should be stated throughout the text (maybe at the end of each chapter). The current conclusions are quite general and since the review is so long, it is not very clear where they come from. I am sure it is quite to for the authors, but it is certainly not the case for the readers. Thus, I would suggest the authors to revise their review and present it in a more readable way before I can recommend it for publication in IJMS.

As already mentioned, oximes are not a distinct family of compounds but rather a versatile functional group that can be easily incorporated into a wide array of chemical frameworks. Given the diverse effects of this functional group on the anticancer properties of resulting molecules, it seemed logical to organize this manuscript by chemical framework. The specific divisions and additional (sub-) chapters suggested by this reviewer is primarily a personal preference that we acknowledge and respect, but might not necessarily result in a better a clearer organization of the manuscript.

As such, we humbly maintain that organizing the manuscript in this manner is the most effective approach, and express our sincere gratitude to this reviewer for all the insightful comments, and hope our responses offer some clarity regarding the raised concerns.